# Interfacial Properties and Structure of Emulsions and Foams Co-Stabilized by Span Emulsifiers of Varying Carbon Chain Lengths and Egg Yolk Granules

**DOI:** 10.3390/foods14010035

**Published:** 2024-12-26

**Authors:** Wenyan Liu, Jingxia Cao, Qixin Zhang, Weiqin Wang, Yuanping Ye, Senwang Zhang, Leiyan Wu

**Affiliations:** 1College of Food Science and Engineering, Jiangxi Agricultural University, Nanchang 330045, China; lwy15679906596@163.com (W.L.); 17741862036@163.com (J.C.); zqx2586512313@163.com (Q.Z.); 2Jiangxi Riyuan Food Co., Shangrao 334604, China; weiwei018@sohu.com (W.W.); yyp13816159004@163.com (Y.Y.); 3Institute of Applied Chemistry, Jiangxi Academy of Sciences, Nanchang 330096, China; 4Jiangxi Zixi Bread Technology Development Co., Fuzhou 335300, China

**Keywords:** egg yolk granules, span, protein structural modifications, interfacial properties, emulsifying agents, foaming stability

## Abstract

Interfacial properties significantly influence emulsifying and foaming stability. We here explore the interfacial behavior of egg yolk granules (EYGs) combined with various Span emulsifiers (Span 20, 40, 60, 80) to assess their solution properties, interface dynamics, and effects on emulsifying and foaming stability. The results unveiled that as the Span concentration increased, particle size decreased from 7028 to 1200 nm, absolute zeta potential increased from 4.86 to 9.26 mv, and the structure became increasingly loosened. This loose structure of EYGs-Span complexes resulted in reduced interfacial tension (γ), higher adsorption rate (Kd), and improved interfacial composite modulus (E) compared with native EYGs. These effects were more pronounced with shorter hydrophobic chain Spans but diminished with longer chain lengths. Enhanced interfacial properties contributed to better emulsification and foaming stability, with EYGs-Span complexes displaying increased emulsifying ability and stability compared with natural EYGs. Emulsifying and foaming stability improved in the order of Span 20 > Span 40 > Span 60 > Span 80 as the Span concentration increased. The correlation analysis (*p* > 0.05) indicated that emulsifying stability was positively associated with interfacial composite modulus and negatively correlated with particle size. Consequently, EYGs-Span composites demonstrate considerable potential for use as effective emulsifiers in food industry applications.

## 1. Introduction

Egg yolk granules (EYGs) primarily comprise high-density lipoproteins (HDLs, 70%), phosvitin (16%), and low-density lipoproteins (LDLs, 12%), interconnected through calcium-phosphorus bridges [1,2]. These attributes render EYGs ideal as emulsifying agents in a diverse range of products [3,4], including mayonnaise, baked goods, cream, and ice cream, particularly in the formulation of mayonnaise, thereby catering to the increasing demand for healthier dietary options [5,6]. However, the compact, calcium-phosphorus bridge structure of EYGs results, reducing emulsification stability in mayonnaise and leads to water-oil separation [7]. Common methods to enhance EYGs emulsification stability include strong alkali treatment or high NaCl ionic strength, which disrupt the compact structure of EYGs, improving their solubility and emulsifying properties. However, these methods risk inducing EYGs coagulation and denaturation, ultimately lowering their nutritional value [8]. Therefore, alternative gentler treatment approaches are needed. According to recent studies, combining emulsifying agents with proteins through electrostatic, hydrogen bonding, or hydrophobic interactions can modify protein structure, enhancing emulsification stability comparably to strong bases and high ionic strengths [9,10]. Additionally, the amphiphilic nature and abundant lipoproteins of EYGs allow them to pair with lipoprotein emulsifiers, which could further improve emulsification stability. Thus, this study examines the combination of Span emulsifiers with EYGs to enhance structural and interfacial properties, aiming to improve emulsification and foaming stability while broadening the application potential.

Span emulsifiers (Span 20, 40, 60, and 80) are nonionic surfactants with identical hydrophilic heads but varying hydrophobic carbon chain lengths. Nonionic emulsifiers, including Span, have been shown to stabilize emulsions more effectively than ionic and amphoteric emulsifiers [11]. However, using a single-component emulsifier, whether macromolecular or small-molecule, poses limitations. While small-molecule emulsifiers quickly adsorb to the interface, the resultant interfacial layer frequently lacks robust spatial repulsion, rendering it inadequate to prevent the coalescence of emulsified droplets [12]. On the other hand, proteins form a highly viscoelastic interfacial network around droplets, enhancing stability, but their adsorption rate at the interface is very slow [8]. Thus, combining proteins with small-molecule emulsifiers offers a synergistic approach to improving the stability of emulsions and foams [13]. Studies have found that soya lecithin induced structural unfolding in EYGs, enhancing flexibility and surface functionality, and ultimately stabilizing emulsions [12]. These findings suggest that the interaction between emulsifiers and proteins can be influenced by the structural characteristics of both the hydrophilic and hydrophobic portions of the emulsifier [10]. However, little is known about how blending EYGs with Span emulsifiers affects protein conformation at the interface or how these structural changes influence emulsification and foaming properties [14].

To enhance food emulsion properties and stability, studying emulsifier-protein interactions is crucial [15]. To address these gaps, this study investigates the effects of different types (Span 20, 40, 60, and 80) and concentrations of Span emulsifiers (0.001%, 0.01%, 0.1%) on the interfacial behavior and structural properties of EYGs. These findings offer insights into optimizing emulsion and foam stability in food applications by leveraging the interactions between EYGs and Span emulsifiers.

## 2. Materials and Methods

### 2.1. Materials

Fresh eggs were purchased from a local supermarket (Nanchang, China) and used within 24 h. Soybean oil was purchased from a local supermarket (Nanchang, China) and stored at room temperature for use. PBS buffer and sodium dodecyl sulfate (SDS) were procured from Solarbio Science & Technology Co., Ltd. (Beijing, China). Additional chemicals, including glycine (Gly), 5,5-dithiobis-(2-nitrobenzoicacid) (DTNB), 8-aniline-1-naphthalenesulfonic acid (ANS), tris(hydroxymethyl)aminomethane (Tris), bromatum kalium (KBr), and Span emulsifiers (Span 20, 40, 60, and 80), were sourced from Aladdin Chemistry Co. (Shanghai, China). Nile red was supplied by Yuanye Biotechnology Co. (Shanghai, China).

### 2.2. Preparation of EYGs

EYGs were prepared following a slightly modified version of a previous method [16]. First, egg yolks were separated from the albumen, and any adhering proteins were removed by rolling the yolks on the filter paper. The yolk membrane was punctured with a sterile needle, and the yolk contents were collected. The yolk was then mixed with PBS buffer (1:1 *w*/*w*, 0.01 mol/L, pH 7.2) and vortexed for 1 h. The mixture was centrifuged at 12,000× *g* for 30 min at 4 °C. The resulting precipitate containing EYGs was collected and lyophilized for further use.

### 2.3. Composite Solution of EYGs with Span Emulsifier

EYGs-Span complex solutions were prepared using a previous method with some modifications [17]. Fresh EYGs were diluted to 10 mg/mL with PBS buffer (0.01 mol/L, pH 7.2) and stirred for 1 h using a magnetic stirrer. Then, the Span emulsifier (0.1%, 0.01%, and 0.001%, *w*/*w*) was added, and the mixture was stirred for another hour. The resulting EYGs-Span complex solutions, summarized in Table 1, were stored at 4 °C for 12 h before use in subsequent experiments.

### 2.4. Zeta Potential and Particle Size

This study followed a previous method with minor modifications [18]. A Malvern Zetasizer Nano (ZS90, Malvern Zetasizer Nano, Great Malvern, UK) was used to measure the zeta potential and particle size of the samples. The solutions were diluted to 0.1 mg/mL with PBS buffer (0.01 mol/L, pH 7.2), and particle size was measured at a 90° scattering angle. All measurements were conducted at 25 °C and repeated three times for accuracy.

### 2.5. Free Sulfhydryl Content

The content of free sulfhydryl groups in the samples was determined using Ellman’s reagent. The sample solutions were mixed with Tris-Gly buffer, followed by the addition of 80 μL of Ellman’s reagent per 2 mL of the solution. The mixture was stirred and incubated in a 37 °C water bath for 30 min. Absorbance was measured at 412 nm, and free sulfhydryl content was calculated as given below [19]:(1)$$\text{-}\mathrm{SH}(\mu \mathrm{moL}/g)=73.53\times A_{412}/C$$
where A_412_ is the sample absorbance at 412 nm, and C is the protein concentration (mg/mL).

### 2.6. Fourier Transform Infrared Spectroscopy

FTIR analysis was performed following a previous method [20]. Lyophilized samples were mixed with KBr (1:100, *v*/*v*) and compressed into slices by using a tablet device. These slices were analyzed using an FTIR spectrometer (IS5, Thermo Fisher, Waltham, MA, USA) at a resolution of 4 cm^−1^ over a wavelength range of 400–4.000 cm^−1^ with 64 scans.

### 2.7. Fluorescence Spectroscopy

EYGs-Span complex solutions were diluted to 0.1 mg/mL with PBS buffer (0.01 mol/L, pH 7.2). Fluorescence spectra of the solutions were recorded at 25 °C by using a fluorescence spectrophotometer (970 CRT, Precision Instrument Co., Shanghai, China) based on a previous method with minor changes [21]. The excitation wavelength was set to 280 nm, with a slit width of 5 nm, high scanning speed, and a scanning range of 300–450 nm. PBS buffer was used as the blank reference.

### 2.8. Ultraviolet Spectrum

Ultraviolet [22] spectra of the solutions (0.1 mg/mL in PBS buffer (0.01 mol/L, pH 7.2)) were recorded using a UV spectrophotometer (SPECORD 200, Analytical Instruments GMBH, Jena, Germany). The spectral range was set to 225–400 nm, following a slightly modified protocol [23].

### 2.9. Surface Hydrophobicity

Surface hydrophobicity was measured as described by Zhao et al. [24], with some modifications. The samples were diluted to 0.1 mg/mL in PBS buffer (0.01 mol/L, pH 7.2) and mixed with 2.5 mM ANS solution at a 1:100 (*v*/*v*) ratio. After 15 min of shaking the mixture in the dark, fluorescence intensity was measured using a spectrophotometer (970 CRT, Precision Instrument Co., Shanghai, China). The parameter settings included an excitation wavelength of 395 nm, high scanning speed, scanning range of 375–425 nm, a slit width of 5 nm, and a sensitivity of 1.

### 2.10. Contact Angle

Lyophilized samples were pressed into tablets with a diameter of 1 cm and a height of 1 mm [25]. A 5 µL droplet of water was applied to the tablet surface using a precision syringe, and the contact angle was measured using an optical video angle meter (OSA100, Lauda Scientific, Ningbo, China). Video recordings over 20 s captured droplet behavior.

### 2.11. Scanning Electron Microscopy

The microstructure of lyophilized samples was examined through scanning electron microscopy (SEM) (GeminiSEM560, ZEISS, Oberkochen, Germany) by following a previous method with some modifications [26]. The powder of the samples was fixed onto aluminum stubs with an adhesive tape, coated with gold, and observed at a magnification of 1000×.

### 2.12. Dynamic Interfacial Tension

The interfacial tension (γ) of the sample solutions was measured utilizing an OSA100 surface analyzer equipped with an automated drop tensiometer, boasting a resolution of 0.01 mN/m. A 10 µL droplet of the sample solution was dispensed into the oil phase, and its shape was immediately recorded with a camera for 2400 s. The interfacial pressure (π) was calculated as follows [20]:(2)$$\pi =\gamma_{0}\;-\;\gamma \;$$
where γ is the interfacial tension of the sample and γ_0_ is the interfacial tension of ultrapure water, γ_0_ = 72.5 ± 0.5 mN/m and the interfacial tension of medium-chain triglycerides, γ_0_ = 22 ± 0.5 mN/m.

### 2.13. Interfacial Adsorption Kinetics

Adsorption rate and kinetics were calculated using the Ward and Tordai equation [27]:(3)$$\pi =2\cdot C_{0}\cdot K\cdot T{\left(\mathrm{Dt}/3.14\right)}^{1/2}$$
where C_0_ denotes the concentration of the emulsifier solution, K represents the Boltzmann constant, D is the diffusion coefficient, T denotes temperature, and t represents the adsorption time.

The kinetics of penetrating and rearranging the sample solutions at the interface can be analyzed using the equation:(4)$$\mathrm{Ln}\left(\left(\pi_{2400}\;-\;\pi_{t}\right)/\left(\pi_{2400}-\;\pi_{0}\right)\right)=\;-K_{i}t$$
where π_2400_, π_0_, and π_t_ represent the interfacial pressure at 2400 s, 0 s, and any second, respectively. K_i_ denotes the diffusion rate.

### 2.14. Interfacial Dilatational Rheology

Interfacial dilatation properties of the sample solutions were analyzed using the OSA100 surface analyzer with parameters of 10 µL droplet volume, 0.1 Hz frequency, 10% amplitude, and 2400-s test duration. The interfacial expansion modulus (E) was calculated as follows [28]:(5)$$E=d\gamma /\left(\mathrm{dA}/A\right)=d\pi /\mathrm{dlnA}$$
(6)$$\pi =\left[\gamma_{0}/\left(A_{a}/A_{0}\right)\right]\left(\cos\theta +\sin\theta \right)$$
where γ_0_ and A_0_ are the interfacial tension and surface area at equilibrium, respectively, A_a_ denotes the expansion stress, and θ represents the phase angle between expansion stress and strain. Interfacial expansion modulus (E) can also be expressed as follows:(7)$$E=E_{d}+iE_{v}\;$$
where E, Ed, and Ev correspond to the composite dilatation modulus, storage modulus, and loss modulus (mN/m), respectively.

### 2.15. Emulsifying Properties

Using the modified version of a previous method [29], emulsifying activity (EA) and stability (ES) indices were determined by mixing EYGs-Span solutions (2:8 *v*/*v*) with soybean oil and homogenizing (homogenizer; T18 digital ULTRA TURRAX, IKA, Breisgau, Germany) the mixture at 12,000 rpm for 3 min. First, 50 µL of the emulsion was dispersed in 5 mL of SDS solution (0.1%, *w*/*v*). Absorbance of this mixture at 500 nm was measured before and after 10 min by using a UV spectrophotometer (SPECORD 200, Analytical Instruments GMBH, Jena, Germany). The EA and ES were calculated as follows:(8)$$\mathrm{EA}=A_{0}$$
(9)$$\mathrm{ES}=\;A_{0}\times 10/\left(A_{0}-A_{10}\right)$$
where 10 is the time of the experimental measurement, A_0_ denotes the initial absorbance (0 min), and A_10_ represents the absorbance after 10 min.

### 2.16. Microstructure of Emulsion Droplets

The microstructure of the EYGs-Span system emulsions was observed using an inverted fluorescence microscope (IX73, Axiolab A, Nankoku City, Japan) following [30] with a few modifications. In total, 10 µL of the emulsion was applied to a slide, covered with a glass slide cover, and photographed under a 40× objective.

### 2.17. Foaming Properties

Foam index tests were performed following with slight modifications [31]. Two milliliters of each EYGs-Span complex solution were homogenized in cylindrical containers (T18 digital ULTRA TURRAX, IKA, Staufen, Germany) at 10,000 rpm for 3 min. At 0 and 30 min, the samples were placed on a horizontal surface and photographed and placed on a horizontal surface using a digital camera. The foam layer of the sample was excavated on a slide for observation, and the microstructure images of the foam were taken under a microscope (Nikon D-Eclipse C1 80i, Melville, NY, USA). In brief, the images of foam were recorded at 40× magnification.

### 2.18. Data Analysis

All experiments were performed in triplicate. Statistical analysis was conducted using SPSS software (v19.0), with one-way analysis of variance and Duncan’s multiple range test. Graphs were prepared using Origin 2021 software.

## 3. Results and Discussion

### 3.1. Average Particle Size and Zeta Potential

As illustrated in Figure 1, the combination of EYGs with Span emulsifiers significantly reduced particle size compared with natural EYGs (7028.67 ± 253.16 nm). Furthermore, the mean particle size of the EYGs-Span complex solutions decreased significantly with the concentration of Span emulsifiers increased (*p* < 0.05), indicating improved dispersibility of the particles. Differences in mean particle sizes were observed based on the type of Span emulsifier used, following the trend: Span 20 < Span 40 < Span 60 < Span 80. These findings suggest that the E-0.1S20 sample (EYGs with 0.1% Span 20) was the most effective at forming small droplets, while E-0.001S80 (EYGs with 0.001% Span 80) was the least effective. A possible explanation is the structural interaction between EYGs, a relatively large globular protein, and Span emulsifiers, which are small-molecule, nonionic surfactants with a low hydrophilic/lipophilic balance. The binding of Span emulsifiers may cause a loosening of the EYG’s protein structure, enhancing EYGs solubility and reducing particle size. Similar phenomena were previously observed when Tween 80 interacted with ovalbumin, as reported by Li et al. [32].

The surface charge on droplets, as reflected by zeta potential, indicates the degree of electrostatic repulsion between particles, which is critical for emulsion stability. Figure 1 shows that the absolute zeta potential of diverse EYGs-Span solutions gradually decreased with increasing emulsifier concentration (*p* < 0.05). All samples exhibited a negative zeta potential, which enhances particle repulsion and prevents droplet coalescence, thereby improving emulsion stability. A significant change was noted in the measured ζ-potential for the solutions: natural EYGs (−4.86 mv) and E-0.1S20 (−9.26 mv), indicating that the Span 20-stabilized emulsion demonstrated superior stability due to its smaller average particle size.

### 3.2. Free Sulfhydryl Group

The free sulfhydryl content can provide insights into the breakage and formation of intermolecular disulfide bonds within protein molecules, as sulfhydryl groups may be exposed on the protein surface or remain buried within its structure. The free sulfhydryl content increased as the concentration of Span emulsifiers increased. (Figure 2; *p* < 0.05) compared to the control (0.263 ± 0.001 μmol/g), with notable differences among the different emulsifiers: Span 20 > Span 40 > Span 60 > Span 80. This trend suggests that as the emulsifier content increases, the interaction with EYGs intensifies, causing the protein structure to unfold and expose previously concealed sulfhydryl groups. The addition of nonionic surfactants is known to influence protein solubilization or hydrophilization [33]. Therefore, the increased free sulfhydryl content may be related to increased EYG solubility.

### 3.3. Secondary Structural Analysis

Protein conformation significantly influences the exposure or burial of various functional groups within the molecule, affecting their accessibility and activity [34]. Consequently, analyzing the conformational changes in EYGs is essential to understanding how emulsifier type and concentration influence the behavior of active groups within the EYGs. Figure 3 displays the spectroscopic analysis of EYGs, highlighting characteristic peaks that elucidate their chemical structure. Peak 1, observed at 3260 cm^−1^, corresponds to hydrogen bonding, a key indicator of interactions within the protein or between the protein and its environment. Peaks 2, 3, and 4 at 2930, 2860, and 1750 cm^−1^, respectively, are associated with lipid components. Additional lipid vibrations were confirmed by peaks 7 and 8 at 1470 and 1380 cm^−1^. Protein-associated signals were evident in peaks 5 and 6 at 1700 and 1600 cm^−1^, indicating amide bonds and protein backbone vibrations. Peaks 9 and 10 at 1240 and 1090 cm^−1^ suggest the presence of phospholipids, crucial components of EYGs. While the overall peak patterns of EYGs-Span complexes closely resemble those of native EYGs, an exception was Peak 1, indicating possible noncovalent interactions between the carbonyl oxygen in the protein and the hydroxyl hydrogen from the emulsifiers. Therefore, this interplay potentially modifies the functional properties of EYGs, enhancing their emulsification performance [10].

Alterations in the amide I band (1600 cm^−1^–1700 cm^−1^) of the FTIR spectrum are widely recognized as markers for changes in the protein secondary structure, making this region particularly informative for assessing such structural modifications [35]. As shown in Table 2, the secondary structure of EYGs predominantly consisted of β-sheets and β-turns, which together accounted for nearly half of the total structure. However, with increasing Span concentration, the proportion of β-sheets decreased, whereas those of random coils, α-helices, and β-turns increased notably. Compared to EYGs, E-0.1S20 exhibited notable variations in β-folding, irregular curling, α-helix content, and β-turn formation, with the effectiveness of Span 20 being superior to that of Span 40, which in turn was more effective than Span 60, and Span 60 outperforming Span 80. This shift suggested that the protein structure became more open and less compact, with interfacial proteins transitioning from an ordered to a more disordered state. Yang et al. [28] deduced that an increase in random coil conformations correlates with enhanced protein flexibility, enabling better adaptability for adsorption at the oil/water interface and promoting superior emulsification. The behavior may be attributed to strengthening van der Waals force interactions between the emulsifier’s hydrophobic moiety and the protein. Additionally, Span emulsifiers, which feature larger head groups compared to other emulsifiers, likely disrupt the protein’s interfacial membrane, inducing conformational alterations in the proteins [10].

### 3.4. Tertiary Structural Changes

Fluorescence spectroscopy is widely used to explore changes in the tertiary structure of proteins, primarily by highlighting variations in the polarity of aromatic amino acids [36]. As shown in Figure 4A, the maximum fluorescence intensity of the EYGs-Span complex increased progressively with higher emulsifier concentrations, compared with that of the natural EYGs (85.45 ± 1.08). The sequence of increase in fluorescence intensity across the samples was as follows: Span 20 > Span 40 > Span 60 > Span 80. Moreover, as the emulsifier concentration increased (Table 3), a slight blue shift was observed in the maximum absorption wavelength of the sample solutions. This shift may be attributed to enhanced interactions between the fluorescent chromophore and water molecules, as well as the disruption of hydrophobic interactions and the elongation of molecular structures upon exposure to Trp/Tyr residues.

UV spectroscopy provides valuable insights into tertiary structural changes in proteins [37]. Figure 4B illustrates that the UV intensity of the EYGs-Span complex augmented with the emulsifier concentration, likely reflecting alterations in the protein’s secondary structure [38]. The UV absorption peak appeared at approximately 280 nm, suggesting that these changes are primarily associated with tyrosine residues. Meanwhile, the absorbance of the sample solutions was higher than that of the control group. At the same concentration, the emulsifiers displayed different absorbance profiles, with the order being: Span 20 > Span 40 > Span 60 > Span 80. These differences may stem from the unfolding and reorganization of EYGs in conjunction with the Span emulsifier at the interface, resulting in enhanced flexibility and increased exposure of chromogenic groups.

Collectively, the results presented in Figure 4A,B indicate that the incorporation of emulsifiers induces a loosening of the EYG structure, with the degree of structural loosening correlating with the emulsifier concentration. Among the four emulsifiers tested, Span 20 demonstrated the most effective interaction with EYGs.

### 3.5. Contact Angle and Surface Hydrophobicity(H_0_)

Effective methods for assessing the hydrophobicity and hydrophilicity of samples include contact angle (θ) and H_0_ measurements [20,32]. The θ value of the natural EYGs was relatively low (19.75° ± 0.5°), indicating their hydrophilic nature (Figure 5). However, as the emulsifier concentration increased, the θ value of the EYGs-Span complex increased, peaking at E-0.1S20 (69.27° ± 0.48°). These results suggest that the EYGs-Span complexes exhibited enhanced hydrophobicity, making them effective stabilizers for emulsion formation, consistent with our previous observations [39]. Additionally, Figure 5 highlights a significant increase in hydrophobicity with increasing emulsifier concentration, following the order: Span 20 > Span 40 > Span 60 > Span 80. The observed higher H_0_ values can be attributed to two factors. First, Span emulsifiers can directly interact with proteins, shielding the hydrophobic regions of the protein or forming micellar complexes with both proteins and surfactants [40]. This interaction leads to changes in the conformation and stability of the protein molecules [41]. Second, alterations in the protein structure may influence the binding capacity of ANS. ANS has a higher affinity for well-structured proteins than unstructured polypeptides [22].

### 3.6. Scanning Electron Microscopy

The sample’s microstructure, including its three-dimensional structure and spatial arrangement, was observed through SEM [42]. The structural changes in the freeze-dried EYGs-Span complex were further illustrated in the SEM images shown in Figure 6. With increasing emulsifier concentration, the fragments of the EYGs-Span complex became smaller and less cohesive compared to the large aggregates in natural EYGs. The structural changes followed this order: Span 20 < Span 40 < Span 60 < Span 80, although Span 80 showed only a minimal effect on the EYGs. This observation is consistent with the variation in the average particle size described in Section 3.1, confirming the hypothesis that the EYG structure becomes looser upon emulsifier addition.

### 3.7. Dynamic Interfacial Tension Analysis

The interfacial tension of EYGs (10 mg/mL), Span (0.001%, 0.01%, and 0.1%), and their mixtures (0.001%, 0.01%, and 0.1%) at the air/water and oil/water (bottom) interfaces was examined (Figure 7). All graphs exhibited an initial sharp decline, attributed to the adsorption of the samples at the interface. This was followed by a leveling off of the curve, reflecting the stabilization of the interface as the protein and emulsifier adsorbed and covered the surface [43]. Compared with natural EYGs, the reduction in interfacial tension caused by the small-molecule emulsifier alone at the oil/water interface was relatively less significant. However, the reduction in interfacial tension in the EYGs-Span composite system followed this order: Span 20 > Span 40 > Span 60 > Span 80. This pattern suggests that an emulsifier’s ability to reduce interfacial tension is strongly influenced by its molecular characteristics, including molecular weight and the number and arrangement of hydrophilic and hydrophobic groups. Span 20, with the lowest molecular weight, experiences less steric hindrance when interacting with egg yolk particles, facilitating quicker adsorption at the interface and greater reduction in interfacial tension [8]. At lower emulsifier concentrations, no significant difference was observed in protein displacement, likely because the protein and low-molecular-weight emulsifiers coexisted at the interface. However, at higher emulsifier concentrations, most proteins were displaced, and the EYGs-Span complex became the dominant interface-active species [11].

In conclusion, the presence of surfactants significantly reduces interfacial tension, with mixtures displaying superior surface activity compared with pure EYGs or Span alone. This suggests a synergistic effect between EYGs and Span in lowering interfacial tension. These interactions promote the rearrangement of the compressed mixed film into a more stable monolayer. Surfactants penetrate the protein films at the interface, facilitating protein displacement and desorption [44]. Similar findings have been reported with mixtures of soy glycinin and soyasaponin at the oil/water interface [45].

### 3.8. Adsorption Kinetics

The protein adsorption process at both the air-water and oil-water interfaces can be divided into three phases: diffusion, infiltration, and rearrangement. During the initial 100 s of protein adsorption, a linear increase in interfacial pressure was observed, with the slope passing through the origin, indicating diffusion-controlled adsorption. The slope of the linear correlation between interfacial pressure (π) and the square root of time (t^1/2^) indicates the initial diffusion rate K*_diff_*. Table 4 summarizes the changes in diffusion, permeation, and rearrangement rates at the interface of the samples. As the concentration of the emulsifiers increased, the K*_diff_* followed the order: Span 20 > Span 40 > Span 60 > Span 80, all of which were greater than the control. Among them, E-0.1S20 showed a significant difference compared with EYGs. The initial adsorption kinetics of proteins and protein particles are typically restricted by their diffusion from the bulk to the interface, which can be influenced by factors such as size, shape, surface charge, and wettability [46]. The slower initial adsorption of E-0.001S can be attributed to the larger size of the formed clusters/aggregates (Figure 1), which hinders rapid diffusion to the interface. By contrast, at higher Span concentrations (0.1% wt), the initial surface tension of the EYGs was markedly reduced, primarily due to the sufficient presence of surfactant materials at the interface during the initial adsorption phase.

After the initial diffusive adsorption, the proteins infiltrate and rearrange at the interface. The penetration rate (K*_p_*) and rearrangement rate (K*_R_*) were calculated using Origin 2021 (Table 4). As evident in the table, the trends of K*_P_* and K*_R_* were inconsistent and lacked a clear pattern, indicating no direct link between the multilayer adsorption of EYGs surrounding the oil droplets and their emulsifying capabilities.

### 3.9. Interface Expansion Rheology Analysis

The surface dilatational modulus (E) is an indicator of macromolecular interactions in interfacial adsorbed proteins. Figure 8 illustrates the changes in the surface dilatational modulus of the interfacial films formed by all EYGs-Span composite systems at the oil/water interface. The E-values of these films increased over time, suggesting the formation of an elastic film at the interface. As the concentration of Span emulsifiers increased, the E-value also increased compared to the pure emulsifier, indicating that the proteins diffused more rapidly to the interface, facilitating the formation of a more robust interfacial film. During the 40-min adsorption period, the adsorption layer displayed viscoelastic behavior, with elasticity being the dominant characteristic. Notably, the E-values for Span concentrations ranging from 0.001% to 0.1% were quite low (3.0–6.0 mN/m), suggesting that pure Span emulsifiers have relatively low interfacial elasticity. These findings align with similar results observed for other low-molecular-weight emulsifier surfactants [47]. It is likely that that small-molecule emulsifiers, due to their size, can penetrate gaps within the protein film and interact with the proteins. This interaction increases the interfacial density, resulting in a more compact and resilient interfacial film.

### 3.10. Emulsifying Activity and Stability

Emulsifying activity refers to the ability to rapidly adsorb onto the surface of oil droplets and facilitate the formation of small droplets during homogenization. Conversely, the stability index measures the emulsion’s resistance to aggregation and phase separation. Compared to the natural EYGs (EA 0.343 m^2^/g, ES 0.452 min), both EA and ES of the EYGs-Span systems increased slightly with increasing emulsifier concentration in the following order: Span 20 > Span 40 > Span 60 > Span 80 (Figure 9). This indicates that the presence of Span emulsifiers boosted the emulsifying activity of the EYGs. Notably, E-0.1S20 exhibited the highest emulsifying activity and stability, which can be attributed to the Span 20’s low molecular weight (346 Da) and minimal steric hindrance. This allowed for more efficient interface adsorption without displacing the proteins [48]. These findings are consistent with those of [49], who reported superior emulsion properties when sodium caseinate was combined with a low-molecular-weight emulsifier, Tween 20, compared to sodium caseinate alone. Furthermore, the dissociation of EYGs may reduce their particle size and release components, such as HDL, which could further augment emulsification [12].

### 3.11. Microstructure Observation of Emulsion

Emulsions, which are small oil droplets dispersed in an aqueous medium, are thermodynamically unstable systems. They are prone to phase separation, flocculation, coalescence, and separation due to gravity. Figure 10 presents the stability of agglomeration in freshly prepared emulsions under an inverted fluorescence microscope. The size and distribution of EYGs significantly improved after being combined with Span emulsifiers, with the following order of effectiveness: Span 20 > Span 40 > Span 60 > Span 80. The effect was more pronounced with increasing concentration, which corresponded with the emulsification results presented in Figure 11. These findings were agreed with the particle size distribution analysis.

### 3.12. Foaming Properties

The foaming properties of EYGs were significantly enhanced by the added Span emulsifiers, with improvements observed as the emulsifier concentration increased (Figure 11a). Moreover, Figure 11b highlights that the presence of Span resulted in smaller, denser foam bubbles, thereby augmenting the overall foam structure of the EYGs. The effectiveness in improving the foam structure followed the order: Span 20 > Span 40 > Span 60 > Span 80. This enhancement is linked to changes in binding affinities and structural modifications of the proteins. The flexibility and openness of a protein’s structure are directly correlated with its foamability [50]. Stronger binding leads to more remarkable structural changes. These findings suggest that the combination of EYGs with Span emulsifiers reduces protein surface tension and loosens the protein structure, thereby facilitating foam formation. Additionally, the higher protein concentration in the EYG fraction further contributed to this effect, as these proteins help stabilize the foam by forming a protective protein layer at the air/water interface [51].

### 3.13. The Possible Mechanisms

Figure 12a presents a heat map for the correlation analysis of the interfacial properties, emulsification characteristics, and protein structural properties of the EYGs-Span composite system. The correlation analysis highlighted significant relationships among interfacial properties, emulsification performance, and protein structural characteristics. Smaller particle sizes are closely linked to higher ES and greater interfacial elasticity, indicating that finer particles contribute to more stable emulsions. The absolute value of zeta potential was positively correlated with both diffusion (K*_diff_*) and penetration rates (K*_p_*), suggesting that stronger electrostatic repulsion improves molecular mobility at the interface. Additionally, EA and ES were positively correlated with the expansion modulus, emphasizing the importance of interfacial elasticity in effective emulsification. Notably, more disordered protein structures, such as irregularly curled conformations, are associated with lower contact angles, indicating improved wettability. Furthermore, the positive correlations between UV and fluorescence spectral maxima and EA suggested that structural rearrangements facilitate augmented emulsification efficiency. These findings illustrate the complex interplay between particle size, interfacial behavior, and protein structure in influencing the stability and functionality of emulsions.

The amphiphilic molecules used in this study share similar chemical structures, meaning that if EYGs interacted only with the hydrophilic groups of the emulsifier, the behavior of EYGs and Span would exhibit slight differences across most tests. However, the primary distinction among Span molecules lies in the length of their hydrophobic alkyl chains (Figure 12b), and this structural variation influences their interaction with EYGs. As the Span concentration increased, the EYGs-Span complex became looser, as reflected by the increase in disordered structures noted in the protein’s secondary structure through FTIR, with similar changes seen in its tertiary structure. At lower concentrations, although the adsorption rate of a small amount of the EYGs-Span complexes improved, even more native EYGs were adsorbed at the interface. However, as the concentration increased, the structure became looser, and the average particle size of the EYGs-Span complexes decreased. This reduces steric hindrance and increases the adsorption rate, leading to augmented adsorption at the interface, thereby improving the dispersion of the emulsion and foam stability. These findings, supported by the dynamic interfacial property’s analysis, suggest that some EYGs interact with the Span’s hydrophobic tails, forming a composite EYGs-Span layer that stabilizes the adsorbed EYGs at the oil/water or air/water interface, possibly as mixed monolayers.

## 4. Conclusions

This study investigated the interaction between EYGs and the nonionic small-molecule surfactant Span at concentrations of 0.001%, 0.01%, and 0.1% relative to the protein content, with a focus on their physicochemical, foaming, interfacial, and structural properties. Compared with the natural EYGs, the addition of Span led to a reduction in the average particle size of the EYGs-Span complex. Additionally, the free sulfhydryl group content, hydrophobicity, and the absolute value of the ζ-potential all increased. At both the air/water and oil/water interfaces, the interfacial tension decreased, whereas the adsorption rate (Kd) and composite modulus (E) increased. These changes contributed to the augmented emulsifying and foaming properties of the EYGs-Span composite system. Moreover, as the Span concentration increased, the binding effect on EYGs followed this order: Span 20 > Span 40 > Span 60 > Span 80. Compared to using strong alkali or high ion concentration, combining span emulsifier with egg yolk particles offers a simpler and more energy-efficient approach, effectively enhancing the emulsification and foaming stability of the EYGs. This study establishes a solid theoretical foundation and technical roadmap for developing specialized EYGs tailored to industries like mayonnaise manufacturing. By integrating Span, it creates high-performance EYGs that reduce costs and elevate product attributes. Looking ahead, the EYGs-Span complex stands as a potent emulsifier, poised to dramatically improve stability, texture, and flavor across various food sectors, including mayonnaise, baked goods, and dairy products. Ultimately, this breakthrough expands the utility of EYGs and ushers in new opportunities for innovative, premium food production.

## Figures and Tables

**Figure 1 foods-14-00035-f001:**
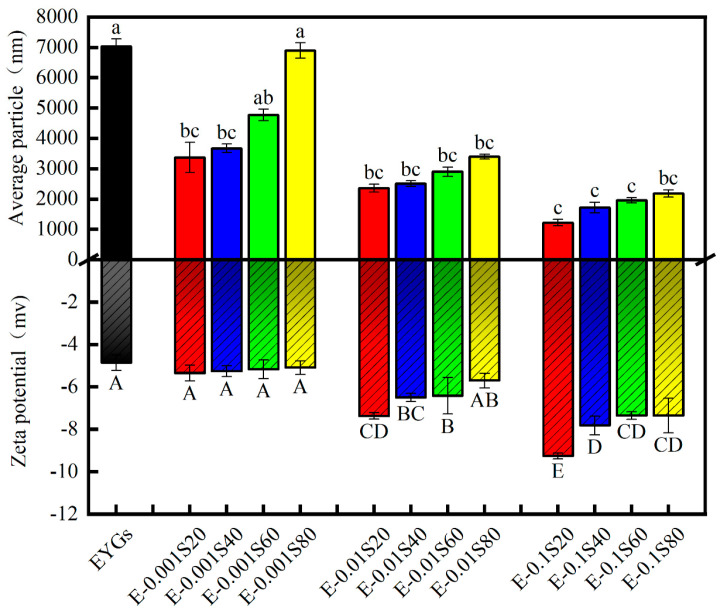
Average particle size and zeta potential of EYGs-Span emulsifiers with different compounding concentrations. Different letters indicate significant differences (*p* < 0.05) in the mean values within the same parameter group.

**Figure 2 foods-14-00035-f002:**
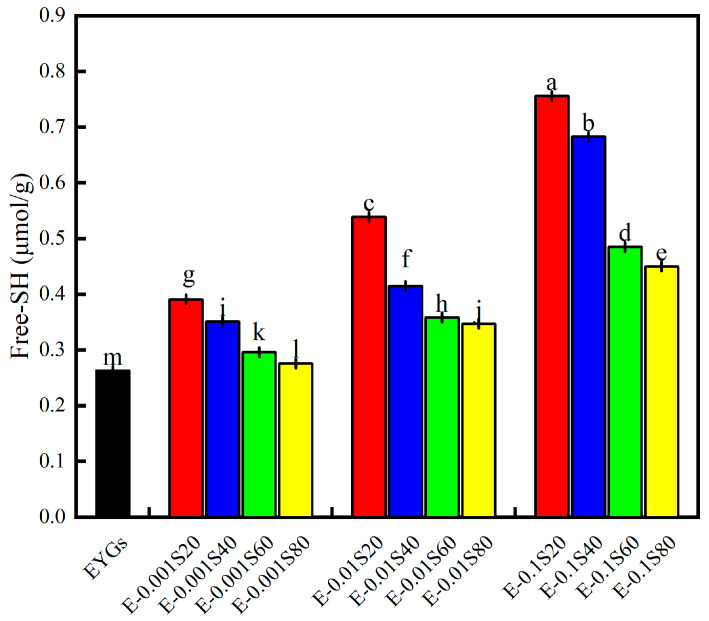
Free sulfhydryl content of EYGs-Span emulsifiers with different compounding concentrations. Different letters indicate significant differences (*p* < 0.05) in the mean values within the same parameter group.

**Figure 3 foods-14-00035-f003:**
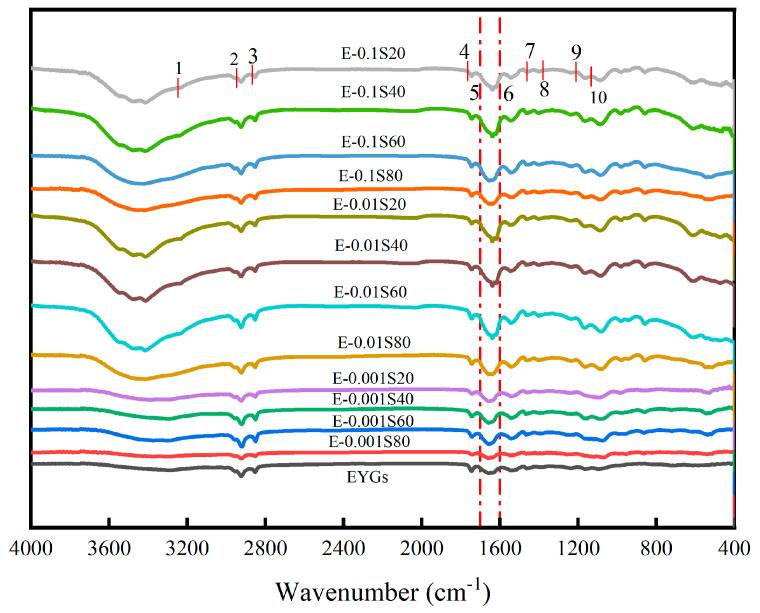
Infrared spectra of EYGs-Span emulsifiers with different compounding concentrations.

**Figure 4 foods-14-00035-f004:**
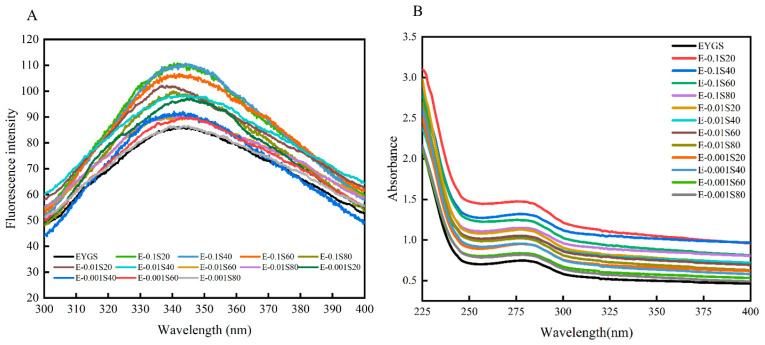
Fluorescence spectra (**A**) and UV spectra (**B**) of EYGs-Span emulsifiers with different compounding concentrations.

**Figure 5 foods-14-00035-f005:**
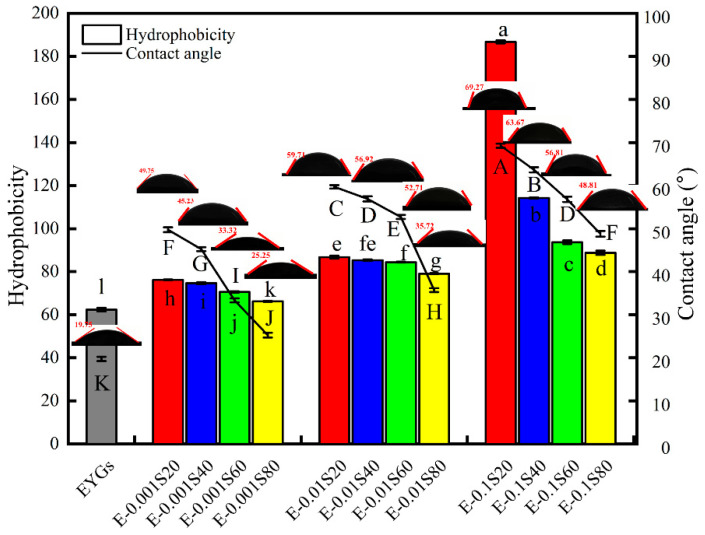
Contact angle and surface hydrophobicity of EYGs-Span composite systems with different compounding concentrations. Different letters indicate significant differences (*p* < 0.05) in the mean values within the same parameter group.

**Figure 6 foods-14-00035-f006:**
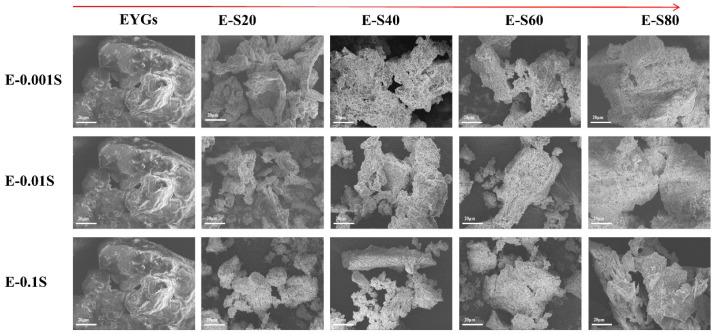
Scanning electron microscopy of EYGs-Span composite systems with different compounding concentrations (magnification 1.00 k×, scale bar is 20 µm).

**Figure 7 foods-14-00035-f007:**
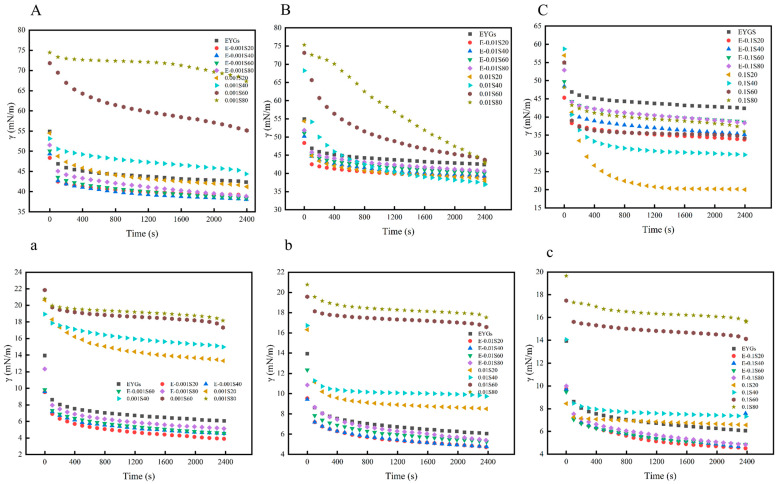
Dynamic interfacial tension versus time at the air/water and oil/water (bottom) interfaces for EYGs-Span emulsifier and Span emulsifier with different compounding concentrations ((**A**,**a**) is 0.001 concentration, (**B**,**b**) is 0.01 concentration, and (**C**,**c**) is 0.1 concentration).

**Figure 8 foods-14-00035-f008:**
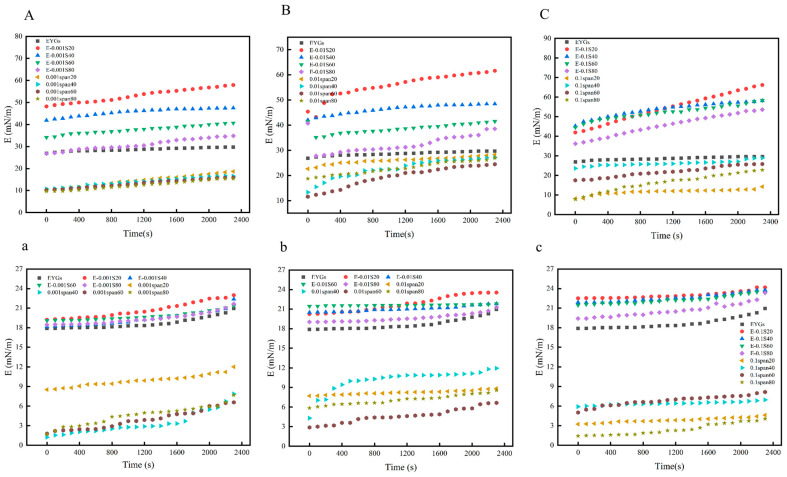
Composite expansion modulus with time for the EYGs-Span emulsifier and Span emulsifier at the air/water and oil/water (bottom) interfaces with different compounding concentrations ((**A**,**a**) is 0.001 concentration, (**B**,**b**) is 0.01 concentration, and (**C**,**c**) is 0.1 concentration).

**Figure 9 foods-14-00035-f009:**
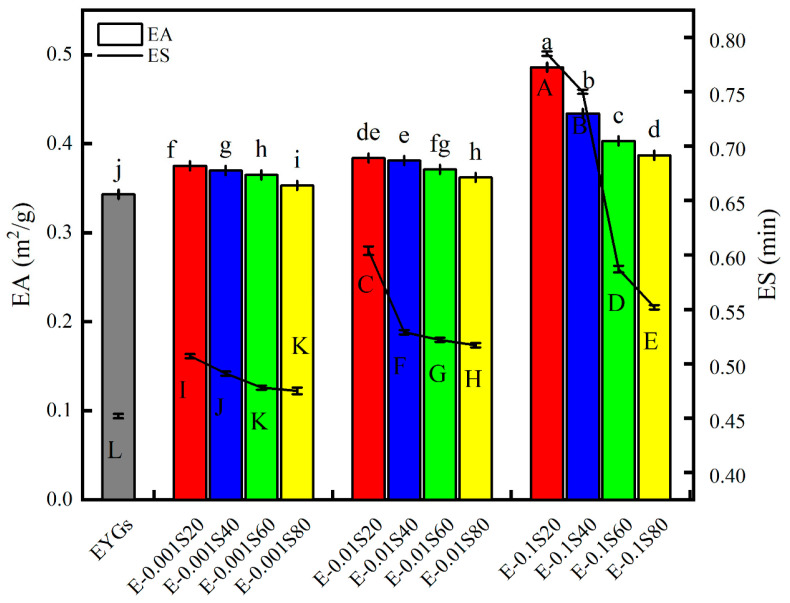
Emulsifying activity and stability of EYGs-Span emulsifiers with different compounding concentrations. Different letters indicate significant differences (*p* < 0.05) in the mean values within the same parameter group.

**Figure 10 foods-14-00035-f010:**
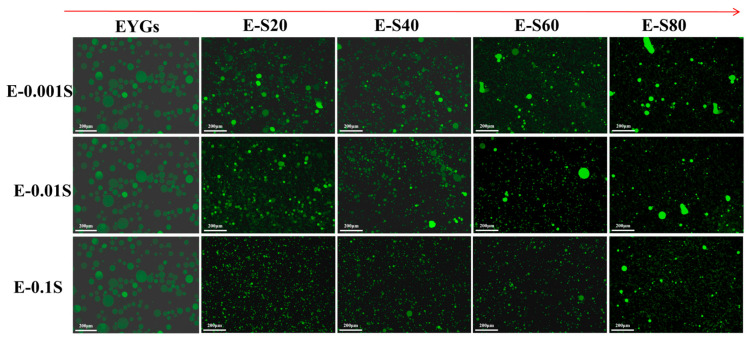
Observation of the emulsion microstructure of EYGs-Span emulsifiers with different compounding concentrations (magnification 40×, scale bar is 200 μm).

**Figure 11 foods-14-00035-f011:**
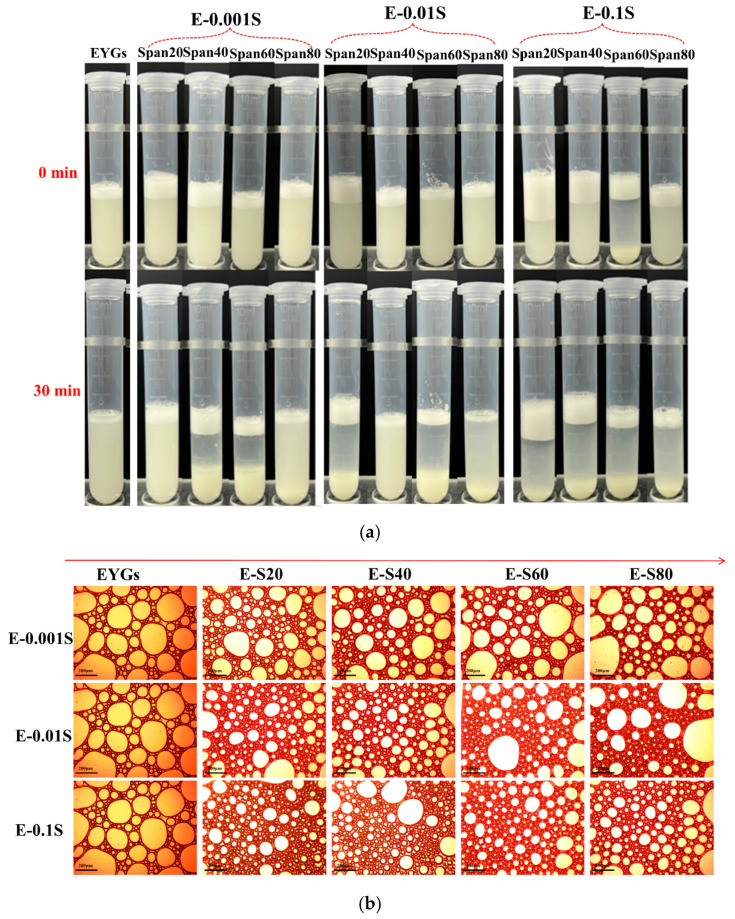
(**a**) Comparison of foaming properties of EYGs-Span emulsifiers with different compounding concentrations at 0 and 30 min. (**b**) Observation of the foam microstructure of EYGs-Span emulsifiers with different compounding concentrations (magnification 40×, scale bar is 200 μm).

**Figure 12 foods-14-00035-f012:**
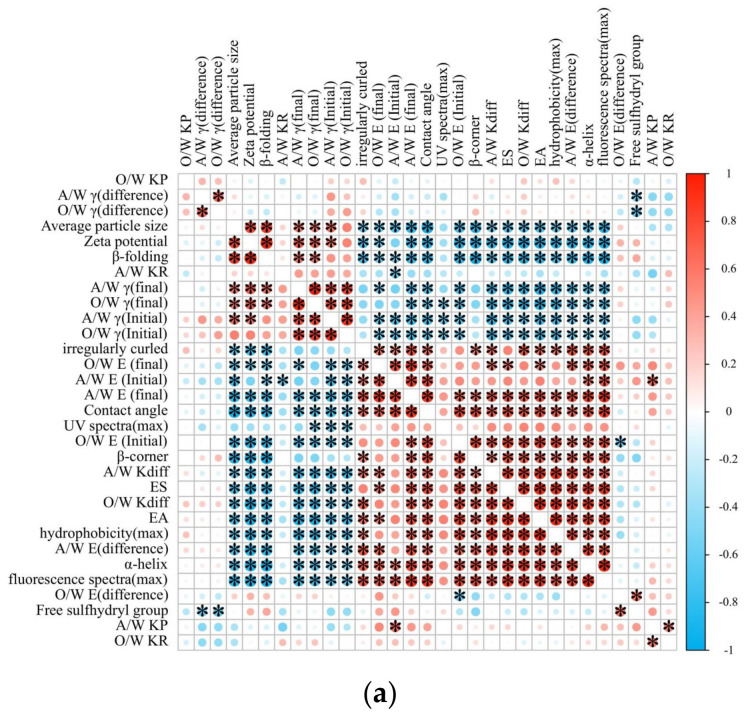
(**a**) The principal component correlation analysis of the EYGs-Span complex, * indicates significant difference (*p* < 0.05). (**b**) Schematic representation of the formation mechanism of emulsions and foams stabilized by EYGs-Span composites.

**Table 1 foods-14-00035-t001:** Coding of each sample with different treatments.

Samples	EYGs Concentration (%, *w*/*w*)	Span Concentration (%, *w*/*w*)
EYGs	1	0
E-0.001S20	1	0.001
E-0.001S40	1	0.001
E-0.001S60	1	0.001
E-0.001S80	1	0.001
E-0.01S20	1	0.01
E-0.01S40	1	0.01
E-0.01S60	1	0.01
E-0.01S80	1	0.01
E-0.1S20	1	0.1
E-0.1S40	1	0.1
E-0.1S60	1	0.1
E-0.1S80	1	0.1

**Table 2 foods-14-00035-t002:** Protein secondary structure content of EYGs-Span with different compounding concentrations.

Samples	β-Folding (%)	Irregular Curling (%)	α-Helix (%)	β-Turning (%)
EYGs	47.61% ± 0.01 ^a^	15.52% ± 0.00 ^bc^	13.17% ± 0.00 ^d^	23.70% ± 0.01 ^e^
E-0.001S20	43.28% ± 0.02 ^bc^	16.73% ± 0.01 ^abc^	14.99% ± 0.01 ^bcd^	25.01% ± 0.01 ^e^
E-0.001S40	45.46% ± 0.02 ^ab^	15.79% ± 0.02 ^abc^	13.75% ± 0.01 ^cd^	25.00% ± 0.01 ^e^
E-0.001S60	45.91% ± 0.00 ^ab^	15.79% ± 0.00 ^abc^	12.99% ± 0.01 ^d^	25.32% ± 0.01 ^e^
E-0.001S80	47.14% ± 0.01 ^a^	15.59% ± 0.00 ^bc^	13.13% ± 0.01 ^d^	24.14% ± 0.01 ^e^
E-0.01S20	30.42% ± 0.01 ^f^	17.78% ± 0.00 ^abc^	18.27% ± 0.00 ^ab^	33.53% ± 0.00 ^bcd^
E-0.01S40	35.37% ± 0.02 ^e^	17.84% ± 0.02 ^abc^	14.77% ± 0.01 ^bcd^	32.02% ± 0.02 ^cd^
E-0.01S60	37.92% ± 0.01 ^de^	14.80% ± 0.03 ^c^	13.09% ± 0.06 ^d^	34.19% ± 0.03 ^bc^
E-0.01S80	40.19% ± 0.02 ^cd^	14.92% ± 0.01 ^c^	14.48% ± 0.02 ^cd^	30.41% ± 0.03 ^d^
E-0.1S20	24.41% ± 0.01 ^h^	18.93% ± 0.00 ^a^	20.63% ± 0.01 ^a^	36.03% ± 0.00 ^ab^
E-0.1S40	26.26% ± 0.02 ^gh^	16.25% ± 0.03 ^abc^	19.86% ± 0.01 ^a^	37.63% ± 0.02 ^a^
E-0.1S60	26.82% ± 0.01 ^gh^	18.32% ± 0.01 ^ab^	19.79% ± 0.00 ^a^	35.07% ± 0.01 ^abc^
E-0.1S80	28.14% ± 0.02 ^fg^	18.68% ± 0.02 ^ab^	17.06% ± 0.02 ^abc^	36.12% ± 0.02 ^abc^

Different letters in the same column in the table indicate significant differences at the *p* < 0.05 level.

**Table 3 foods-14-00035-t003:** Changes in the fluorescence spectra of EYGs-Span with different compounding concentrations.

Samples	Fluorescence Spectra (max)	λ_max_ (nm)
EYGS	85.45 ± 1.08 ^h^	344.90 ± 0.36 ^a^
E-0.001S20	97.37 ± 0.74 ^e^	344.17 ± 0.40 ^a^
E-0.001S40	92.43 ± 0.92 ^f^	339.67 ± 0.80 ^d^
E-0.001S60	89.18 ± 0.66 ^g^	341.83 ± 0.65 ^b^
E-0.001S80	86.40 ± 0.91 ^h^	344.20 ± 0.56 ^a^
E-0.01S20	101.79 ± 0.43 ^c^	337.90 ± 0.30 ^e^
E-0.01S40	97.77 ± 0.90 ^ed^	342.10 ± 0.30 ^b^
E-0.01S60	89.63 ± 0.82 ^g^	344.17 ± 0.35 ^a^
E-0.01S80	89.78 ± 0.97 ^g^	338.20 ± 0.40 ^e^
E-0.1S20	110.25 ± 0.63 ^a^	340.67 ± 0.35 ^c^
E-0.1S40	109.03 ± 0.46 ^a^	344.60 ± 0.36 ^a^
E-0.1S60	104.99 ± 0.63 ^b^	342.23 ± 0.35 ^b^
E-0.1S80	99.00 ± 0.87 ^d^	339.90 ± 0.36 ^dc^

Different letters in the same column in the table indicate significant differences at the *p* < 0.05 level.

**Table 4 foods-14-00035-t004:** Adsorption kinetic parameters of EYGs-Span emulsifiers with different compounding concentrations at the air/water and oil/water interfaces.

Samples	Air/Water Interface			Oil/Water Interface		
	K_diff_ (mN/m^−1^∙s^−0.5^)	K_p_ × 10^3^ (s^−1^)	K_r_ × 10^3^ (s^−1^)	K_diff_ (mN/m^−1^∙s^−0.5^)	K_p_ × 10^3^ (s^−1^)	K_r_ × 10^3^ (s^−1^)
EYGs	0.444 ± 0.002 ^h^	−1.107 ± 0.025 ^e^	−2.710 ± 0.114 ^c^	0.176 ± 0.001 ^k^	−1.037 ± 0.025 ^a^	−2.633±0.129 ^cde^
E-0.001S20	0.507 ± 0.003 ^f^	−0.930 ± 0.01 ^b^	−2.997 ± 0.095 ^d^	0.202 ± 0.001 ^g^	−1.183 ± 0.031 ^c^	−2.367 ± 0.112 ^abc^
E-0.001S40	0.508 ± 0.003 ^f^	−1.083 ± 0.021 ^de^	−3.090 ± 0.079 ^d^	0.196 ± 0.001 ^h^	−1.193 ± 0.025 ^cd^	−2.703 ± 0.179 ^de^
E-0.001S60	0.507 ± 0.002 ^f^	−0.920 ± 0.01 ^b^	−2.987 ± 0.095 ^d^	0.188 ± 0.001 ^i^	−1.117 ± 0.025 ^b^	−2.450 ± 0.147 ^bcd^
E-0.001S80	0.509 ± 0.004 ^f^	−1.313 ± 0.012 ^g^	−2.170 ± 0.118 ^a^	0.181 ± 0.001 ^j^	−1.253 ± 0.021 ^e^	−2.920 ± 0.087 ^e^
E-0.01S20	0.643 ± 0.003 ^b^	−0.813 ± 0.04 ^a^	−2.560 ± 0.044 ^c^	0.245 ± 0.002 ^e^	−1.203 ± 0.035 ^cde^	−2.127 ± 0.096 ^a^
E-0.01S40	0.500 ± 0.002 ^g^	−1.037 ± 0.015 ^cd^	−2.907 ± 0.091 ^d^	0.220 ± 0.001 ^f^	−1.157 ± 0.021 ^bc^	−2.387 ± 0.09 ^abcd^
E-0.01S60	0.511 ± 0.002 ^ef^	−1.073 ± 0.021 ^de^	−2.527 ± 0.067 ^bc^	0.205 ± 0.001 ^g^	−1.163 ± 0.021 ^bc^	−2.673 ± 0.189 ^cde^
E-0.01S80	0.500 ± 0.002 ^g^	−1.207 ± 0.015 ^f^	−2.360 ± 0.044 ^b^	0.204 ± 0.001 ^g^	−1.210 ± 0.03 ^cde^	−2.137 ± 0.175 ^ab^
E-0.1S20	0.827 ± 0.004 ^a^	−1.113 ± 0.031 ^e^	−2.960 ± 0.181 ^d^	0.501 ± 0.004 ^a^	−1.043 ± 0.021 ^a^	−2.663 ± 0.189 ^cde^
E-0.1S40	0.589 ± 0.003 ^c^	−0.990 ± 0.026 ^c^	−3.053 ± 0.095 ^d^	0.345 ± 0.002 ^b^	−1.253 ± 0.021 ^e^	−2.580 ± 0.236 ^cd^
E-0.1S60	0.576 ± 0.003 ^d^	−1.080 ± 0.02 ^de^	−2.557 ± 0.084 ^c^	0.326 ± 0.003 ^c^	−1.243 ± 0.05 ^de^	−2.510 ± 0.147 ^cd^
E-0.1S80	0.517 ± 0.003 ^e^	−1.193 ± 0.04 ^f^	−2.560 ± 0.061 ^c^	0.317 ± 0.003 ^d^	−1.043 ± 0.021 ^a^	−2.663 ± 0.189 ^cde^

Different letters in the same column in the table indicate significant differences at the *p* < 0.05 level.

## Data Availability

The original contributions presented in this study are included in the article. Further inquiries can be directed to the corresponding authors.

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
