# Peer review of "Interfacial Properties and Structure of Emulsions and Foams Co-Stabilized by Span Emulsifiers of Varying Carbon Chain Lengths and Egg Yolk Granules"

_foods, 2024, doi:10.3390/foods14010035_

Round 1
Reviewer 1 Report
Comments and Suggestions for Authors
Dear authors,
The study investigates the interfacial behaviour of egg yolk granules (EYGs) in combination with different Span emulsifiers (Span20, 40, 60, 80) and focuses on their solution properties, interfacial dynamics and their influence on the stability of emulsions and foams.
The topic is relevant and original. It investigates the use of egg yolk granules as a natural emulsifier in combination with non-ionic surfactants to improve the stability of emulsion systems. The study proposes gentler methods to optimise interfacial stability and offers an alternative to conventional chemical or physical treatments.
The research provides valuable data on the interaction between surfactants with different hydrophobic chain lengths and proteins in the EYG and gives a detailed insight into the structural changes and behaviour at the interface. The results are useful for the development of natural emulsifiers in the food industry.
Some aspects of the methodology should be improved:
- L155-L156 - The exact resolution of the OSA100 analyser used in the dynamic tests is not specified
- L156-157 - It is necessary to mention the frequency of sample recording
- L409 - The name of the software used, the producer and the country are not mentioned.
- It would have been helpful to use alternative surfactants (e.g. Tween) for comparison.
- L296, L422, L-530 - Statistically significant values highlighted in the tables require further explanation
- The references should be supplemented by references to the practical application of these compounds in food.
- Other examples of emulsifiers used in the food industry for the emulsification of EYGs should be added;
- Results of the analysis of the long-term stability of the combinations (e.g. phase separation over time or degradation of functional properties) should be added, if available;
- A comparison between EYGs-Span and other existing solutions should be presented in the conclusions to emphasise its superiority;
- The conclusions should also indicate future research directions.
As previously mentioned I recommend a minor revision of this article.
Research contributes significantly to the understanding of natural emulsifiers and their potential to improve emulsion-based systems and is therefore a valuable resource for the field.
Author Response
-
L155-L156 - The exact resolution of the OSA100 analyser used in the dynamic tests is not specified.
Response from the Authors
We sincerely appreciate the reviewer's comment. Regarding the dynamic tests, the OSA100 analyzer utilized had a resolution of 0.01 mN/m. Taking the reviewer's valuable suggestion into account, I have incorporated the specific resolutions into the revised manuscript on lines 155-157.
- L156-157 - It is necessary to mention the frequency of sample recording.
Response from the Authors
Thank you very much for your suggestion. The OSA 100 analyzer is assessed dynamic interfacial tension and interfacial expansion rheology. Specifically, for interfacial expansion rheology, it operates at a specified frequency, acquiring the composite modulus (E) through periodic droplet oscillations. On the other hand, dynamic interfacial tension(γ) is calculated by utilizing the shape of the droplet and applying the Young-Laplace equation, without requiring the setting of a particular frequency.
- L409 - The name of the software used, the producer and the country are not mentioned.
Response from the Authors
Thank you for your reminder. The software in question is Origin 2021, produced by Origin Lab in the USA. As per the reviewer's suggestion, mention of this specific software has been incorporated into the revised manuscript on line 409.
- It would have been helpful to use alternative surfactants (e.g. Tween) for comparison.
Response from the Authors
We express our sincere appreciation to the reviewer for this comment. The abundance of lipoproteins in EYGs naturally predisposes them to effectively pair with lipophilic Span emulsifiers, thereby bolstering the emulsification stability. In light of this, we made a deliberate choice to employ lipophilic Span emulsifiers for binding with EYGs, as preliminary tests with other emulsifiers, including hydrophilic ones, did not yield comparable results. Notably, within our research group, there are ongoing studies focusing on the interaction between hydrophilic proteins and Tween emulsifiers. This provides an opportunity for future work to conduct comparative analyses, leveraging the expertise and findings from these parallel investigations.
- L296, L422, L-530 - Statistically significant values highlighted in the tables require further explanation
Response from the Authors
Thanks for the feedback. We have included additional explanations of significant values in the revised manuscript (lines 285-288 and 405-406). On line 530, instead of presenting a table, Figure 12.1 is provided, which contains a correlation analysis. Based on this insight, appropriate modifications have been made for the notable discrepancies discussed on lines 500-515.
- The references should be supplemented by references to the practical application of these compounds in food.
Response from the Authors
Thank you for your suggestion. As suggested by the reviewer, the authors have researched and added more literatures to support the practical application of these compounds in food, and they are referred as [2], [4], [5], [6] and [7] in the revised manuscript, as seen in lines 583-597.
- Other examples of emulsifiers used in the food industry for the emulsification of EYGs should be added;
Response from the Authors
We would like to express our sincere gratitude to the reviewer for providing the comment. In accordance with the reviewer's suggestion, we have incorporated additional examples of the use of EYGs in the food industry for emulsification, namely mayonnaise, baked goods, cream, and ice cream, in the revised manuscript. These additions can be found on lines 35-38.
- Results of the analysis of the long-term stability of the combinations (e.g. phase separation over time or degradation of functional properties) should be added, if available;
Response from the Authors
We are deeply grateful for your invaluable positive feedback and constructive suggestions. The authors fully acknowledge the reviewer's perspective that incorporating additional experimental results would significantly deepen our comprehension of the long-term stability of the combinations. It is important to note that the current manuscript follows the prevalent method for assessing emulsification stability, which typically includes a verification period of 10 minutes, as referenced in line 458. We appreciate your suggestion, and fully consider this metric in future applications and to dedicating further efforts towards refining and enhancing our experimental methodologies.
- A comparison between EYGs-Span and other existing solutions should be presented in the conclusions to emphasise its superiority;
Response from the Authors
We appreciate the thoughtful review and constructive feedback provided by reviewer. As suggested by the reviewer, the comparison between EYGs-Span and other existing solutions has been added in the conclusions of revised manuscript on lines 557-560.
- The conclusions should also indicate future research directions.
Response from the Authors
Thank you for the helpful comment. As suggested by the reviewer, the future research directions have been added in the conclusions of revised manuscript on lines 560-567.

Reviewer 2 Report
Comments and Suggestions for Authors
The paper entitled "Interfacial properties and structure of emulsions and foams co-stabilized by Span emulsifiers of varying carbon chain lengths and egg yolk granules“presents the combination of egg yolk granules (EYGs) with various Span (Span20, 40, 60, 80) as emulsifiers and foaming stabilizer.
I suggest the work should be accepted with major corrections. I would like to leave some comments to support my decision:
1. The paper is well written, the length is appropriate, the introduction is well organized.
2. The hypothesis and main idea should be always raised throughout the whole manuscript.
3. In the introduction, include a reference in line 68 (a review, preferentially) to state its significance.
4. Could you clarify the source of the calculations for Table 2 and explain how the FTIR spectra contribute to deriving them? Please avoid zero errors in Table 2.
5. Line 313, why tryptophan is not considered as a contributor in fluorescence and absorption spectra? How might this impact the results and their interpretation? Please clarify.
6. Figure 4: The absorbance spectra show a high baseline at longer wavelengths and significant solvent contribution at short wavelengths. Could you provide the corrected spectra? The spectrum of E-0.1S20 appears to have a high dispersive contribution, which should be corrected to accurately assess the absorbance maximum shift.
7. Table 3: How was the position of the maximum determined? Please explain the mathematical analysis applied. The uncertainty appears to be extremely low. Was the Full Width at Half Maximum (FWHM) used to determine maximum position? Was the fitting performed in terms of energy instead of wavelengths?
8. Please include the method / equation used to calculate surface hydrophobicity (H₀) and raw spectra of ANS in the mixtures.
9. The author’s claim that based on the results presented in Figure 7, it confirms the hypothesis that the EYG structure becomes looser upon emulsifier addition. However, the particles range from 3 to 7 microns and cannot be observed at the magnification shown. Lyophilization may result in non-reproducible structures and alter the particles. Please support your statement with the appropriate references.
10. Why particle size is larger in fluorescence images presented in figure 10, 11.2 than in figure 1?
11. What is the optimal formulation suggested by the authors for its application in mayonnaise?
12. The schematic representation in Figure 12.2 is a good attempt to summarize the work but it is not fully understood. Is it applicable to all of Spans? Please clarify this aspect so that it remains in harmony with lines 512-528, in which it is mentioned that structural variation of Spans influences their interaction with EYGs.
Please be more explanatory and incisive in the relationship between the figures 12.1 y 12.2.
13. Include a scale bar in the attached optical images.
14. Correct Equation 3.
15. Line 161: Rewrite the sentence if it refers to the subtraction relative to the organic phase or water. Review the units; the quotient should not have any.
16. In line 197, correct the typo and explain the meaning and equation of ΔT.
17. Please clarify which samples exhibit significant differences in their mean values (a, b, c, d).
18. The Span concentrations in Table 1 do not match the x-axis in Figure 1.
19. Review the English language, as there are some typographical errors.
20. Line 231: Include the cited author’s name.
21. Line 266 and others: Correct the unit to cm⁻¹.
21. Lines 69-70 refine the text.
23. Line 97 include the RPM.
24. Line 142 revise excitation wavelength and emission range.
25. Line 145: include all dimensions.
26. Use a period (.) as the decimal separator instead of a comma (,) for thousands (line 121).
Author Response
- The paper is well written, the length is appropriate, the introduction is well organized.
Response from the Authors
We greatly appreciate your concise summary of the manuscript and your encouraging commentary. We would like to express our sincere gratitude for your diligent and professional review work, as well as for your constructive comments and invaluable suggestions.
- The hypothesis and main idea should be always raised throughout the whole manuscript.
Response from the Authors
Thank you for your constructive feedback. We appreciate the importance of maintaining clarity and consistency in presenting our hypothesis and main idea throughout the manuscript. In response to your comments, we have revised the text on lines 35-37, 57-59 and 558-568 to ensure that the hypothesis and main idea are not only stated clearly but also reinforced in each section, helping readers to follow the logical flow of our arguments and findings. We believe that this will greatly enhance the readability and coherence of our work. Thank you once again for your valuable input.
- In the introduction, include a reference in line 68 (a review, preferentially) to state its significance.
Response from the Authors
Thank you very much for your thoughtful consideration and valuable feedback on our manuscript. The authors fully concur with the reviewer's suggestion regarding the inclusion of a reference that underscores its significance. In the revised manuscript, it has been added as the reference [14], as seen in line 611, meanwhile the other related references have also been referred.
- Could you clarify the source of the calculations for Table 2 and explain how the FTIR spectra contribute to deriving them? Please avoid zero errors in Table 2.
Response from the Authors
Thanks for kindly reminding us to clarify this point. Table 2 was calculated by analyzing the amide I band (1600 cm−1–1700 cm−1) of the FTIR spectrum using the Peak fit software. The calculation yielded a standard error of zero, a result stemming from the precision maintained by retaining significant digits to two decimal places. Then allows us to present data that does not suffer from rounding errors associated with truncating to fewer decimal places, thereby demonstrating the reliability of our data.
Table 2. Protein secondary structure content of EYGs-Span with different compounding concentrations
|
Samples |
β-folding (%) |
irregular curling (%) |
α-helix (%) |
β-turning (%) |
|
EYGs |
47.61%±0.005a |
15.52%±0.002bc |
13.17%±0.002d |
23.70%±0.010e |
|
E-0.001S20 |
43.28%±0.024bc |
16.73%±0.012abc |
14.99%±0.008bcd |
25.01%±0.005e |
|
E-0.001S40 |
45.46%±0.022ab |
15.79%±0.016abc |
13.75%±0.009cd |
25.00%±0.014e |
|
E-0.001S60 |
45.91%±0.002ab |
15.79%±0.001abc |
12.99%±0.013d |
25.32%±0.012e |
|
E-0.001S80 |
47.14%±0.013a |
15.59%±0.003bc |
13.13%±0.012d |
24.14%±0.012e |
|
E-0.01S20 |
30.42%±0.008f |
17.78%±0.004abc |
18.27%±0.003ab |
33.53%±0.003bcd |
|
E-0.01S40 |
35.37%±0.020e |
17.84%±0.015abc |
14.77%±0.014bcd |
32.02%±0.019cd |
|
E-0.01S60 |
37.92%±0.007de |
14.80%±0.028c |
13.09%±0.055d |
34.19%±0.031bc |
|
E-0.01S80 |
40.19%±0.018cd |
14.92%±0.008c |
14.48%±0.017cd |
30.41%±0.029d |
|
E-0.1S20 |
24.41%±0.012h |
18.93%±0.004a |
20.63%±0.009a |
36.03%±0.001ab |
|
E-0.1S40 |
26.26%±0.024gh |
16.25%±0.031abc |
19.86%±0.012a |
37.63%±0.015a |
|
E-0.1S60 |
26.82%±0.011gh |
18.32%±0.013ab |
19.79%±0.001a |
35.07%±0.005abc |
|
E-0.1S80 |
28.14%±0.024fg |
18.68%±0.020ab |
17.06%±0.015abc |
36.12%±0.019abc |
Different letters in the same column in the table indicate significant differences at the P < 0.05 level
- Line 313, why tryptophan is not considered as a contributor in fluorescence and absorption spectra? How might this impact the results and their interpretation? Please clarify.
Response from the Authors
We appreciate the opportunity to provide further clarification on this important point. Several studies have demonstrated that alterations in the tertiary structure of proteins result in the exposure of internal hydrophobic groups and a greater degree of protein unfolding, consequently enhancing the fluorescence intensity of aromatic amino acids[1]. Given that the findings in this study indicate an elevation in surface hydrophobicity (H0) and contact angle, suggesting an increase in hydrophobicity, it is plausible to conclude that the observed augmentation in fluorescence intensity is associated with the unfolding of the protein structure. Furthermore, we concur that tryptophan also contributes to this increase in fluorescence intensity, as mentioned in line 317 of the manuscript.
- Figure 4: The absorbance spectra show a high baseline at longer wavelengths and significant solvent contribution at short wavelengths. Could you provide the corrected spectra? The spectrum of E-0.1S20 appears to have a high dispersive contribution, which should be corrected to accurately assess the absorbance maximum shift.
Response from the Authors
We are profoundly thankful for your expert and thorough review of our article. In response to your concerns, we have conducted the necessary baseline correction and incorporated the corrected spectra into the article. It is noteworthy that, upon performing this correction, no substantial differences were observed between the corrected and uncorrected spectra for this particular instance. Nevertheless, we attach great importance to your suggestion and will handle such issues with greater caution in our future work, continually refining our methods and processes to ensure the accuracy and reliability of our research.
Figure 4. fluorescence spectra (A)and UV spectra (B) of EYGs-Span emulsifiers with different compounding concentrations.
- Table 3: How was the position of the maximum determined? Please explain the mathematical analysis applied. The uncertainty appears to be extremely low. Was the Full Width at Half Maximum (FWHM) used to determine maximum position? Was the fitting performed in terms of energy instead of wavelengths?
Response from the Authors
Thank you for your professional review work on our manuscript. To determine the maximum peak in Table 3, the process involves several distinct steps. Firstly, an endogenous fluorescence spectrum is plotted through the Origin. Subsequently, this spectrum undergoes peak analysis for fitting purposes. During this analysis, peaks and their corresponding wavelengths are identified. Finally, the identified peaks are sorted to ascertain the maximum peak and its associated wavelength.
Previously, due to my lack of understanding, we did not use the Full Width at Half Maximum (FWHM) method to determine the maximum position. Now that we are aware of its significance, we are unable to employ it for verification due to time constraints during the revision process. However, we will definitely consider applying it in our future work.
Initially, because after reviewing a substantial amount of literature, we discovered that this data is predominantly represented in relation to wavelengths, so we hadn't considered performing the fitting using energy instead of wavelengths. Furthermore, due to the limitations of the available instrumentation, the relationship can solely be displayed in terms of wavelengths alongside their corresponding fluorescence intensities or UV absorptions. Although there is an inverse relationship between energy and wavelength that could theoretically be utilized through calculation, the current equipment limitations prevent us from explicitly showing this relationship.
- Please include the method / equation used to calculate surface hydrophobicity (H₀) and raw spectra of ANS in the mixtures.
Response from the Authors
We thank the reviewer for raising these important points. Typically, the surface hydrophobicity index (H0) is derived through linear regression analysis of the curve illustrating the relationship between fluorescence intensity and protein concentration, with the slope of this line serving as H0. However, in the context of this paper, where protein concentrations are consistent and where contact angle data has been integrated with surface hydrophobicity data, the calculation of H0 has been simplified. Consequently, we have opted to approximate H0 using the maximum fluorescence intensity observed, rather than relying on spectral graphs typically represented by the maximum of fluorescence intensity.
In the current study, the utilization of fluorescence spectrograms with ANS was omitted, as the surface hydrophobicity index (H0) was integrated with contact angle data. However, upon request, these spectrograms can be furnished as supplementary information to provide further clarification in the reply.
Figure. The raw spectra of ANS in the mixtures
- The author’s claim that based on the results presented in Figure 7, it confirms the hypothesis that the EYG structure becomes looser upon emulsifier addition. However, the particles range from 3 to 7 microns and cannot be observed at the magnification shown. Lyophilization may result in non-reproducible structures and alter the particles. Please support your statement with the appropriate references.
Response from the Authors
We thank the reviewer for pointing out this potential confound. As demonstrated in our current study, we measured the average particle size of freshly prepared egg yolk particles, which were proportionally mixed with Span emulsifier and subsequently diluted tenfold with PBS solution. The obtained average particle size ranged from 1 to 7 microns. Additionally, we employed scanning electron microscopy to directly assess the lyophilized powder of the EYGs-Span complex. It is plausible that the discrepancies in results stem from variations between the sample preparations. Certainly, we will give your invaluable suggestions serious consideration. In parallel, we will include the original picture of SEM in the attachment for a more thorough examination and to facilitate further enhancements in our subsequent work.
- Why particle size is larger in fluorescence images presented in figure 10, 11.2 than in figure 1?
Response from the Authors
We are deeply appreciative of the chance to offer further clarification on this significant point. In our research, particle size measurements were conducted using solutions that comprised either egg yolk particles (EYGs) alone or a complex of EYGs with the Span emulsifier. However, it is important to note that the optical microscopic observations presented in Figure 10 pertain to an emulsion, where the dispersion of oil droplets has a notable impact on the particle size measurement, causing it to appear larger than it actually is. Furthermore, Figure 11.2 showcases optical microscopic observations of the foam after its formation. Consequently, the actual quantity measured in this context is the foam's volume, which results in a particle size measurement that is less than the dimensions of both the emulsion and the foam.
- What is the optimal formulation suggested by the authors for its application in mayonnaise?
Response from the Authors
Thank you to the reviewer for posing such a practical question. Based on a thorough comparison of experimental results, it has been determined that the optimal compound emulsifier is E-0.1S20. Emulsifier is typically incorporated into mayonnaise at a concentration of 0.5% relative to the total raw material content. However, due to the complexity of the mixing system after the addition of emulsifier, the specific addition ratio needs to be determined by further experiments.
- The schematic representation in Figure 12.2 is a good attempt to summarize the work but it is not fully understood. Is it applicable to all of Spans? Please clarify this aspect so that it remains in harmony with lines 512-528, in which it is mentioned that structural variation of Spans influences their interaction with EYGs.
Please be more explanatory and incisive in the relationship between the figures 12.1 y 12.2.
Response from the Authors
We are grateful for the chance to offer additional clarification on this crucial matter. As we can see in the Figure 12.2, at lower concentrations, while a slight improvement in the adsorption rate of a minor portion of the EYGs-Span complexes was observed, a greater number of native EYGs were adsorbed at the interface. However, as the concentration increased, the structure loosened, resulting in a decrease in the average particle size of the EYGs-Span complexes. The shorter the carbon chain of these complexes, the looser the structure of EYGs and the smaller their average particle size, which enhances the diffusion rate at the interface. Meanwhile, the longer the carbon chain, the weaker the binding effect becomes. Consequently, this reduction in steric hindrance subsequently enhances the adsorption rate, leading to increased adsorption at the interface, which in turn improves emulsion dispersion and foam stability. The above conclusions are based on the results of interfacial adsorption kinetics analysis, scanning electron microscopy, average particle size, protein secondary and tertiary structure analysis and other experimental results.
Based on the comprehensive analysis of all the data presented in this study, the mechanism of Span 20, 40, 60, and 80 aligns with the one depicted in Figure 12.2. However, further verification is required to determine whether all Span emulsifiers conform to this mechanism.
Figure 12.1 presents a correlation analysis that elucidates the relationships among key influencing factors, including average particle size, diffusion (Kdiff), and protein structure, among others. This analysis aims to provide insights that can further refine and enhance the understanding of the mechanism outlined in Figure 12.2.
- Include a scale bar in the attached optical images.
Response from the Authors
Thank you to the reviewer for their meticulous and comprehensive feedback. In response, we have improved the visibility of the scale in the picture to ensure greater clarity. Furthermore, we have incorporated a note in Figure 10, clearly indicating the dimensions of the scale both visually within the graphic and in the accompanying caption for better understanding. Additionally, we will upload the original picture in the attachment for further review and analysis.
- Correct Equation 3.
Response from the Authors
We thank the reviewer for bringing this to our attention. The authors are sorry for our careless mistakes. As suggested by the reviewer, the Equation 3 has been corrected in the revised manuscript, as seen in line 164.
- Line 161: Rewrite the sentence if it refers to the subtraction relative to the organic phase or water. Review the units; the quotient should not have any.
Response from the Authors
The authors apologize for confusing the reviewer that the formulation of the interfacial tension of ultrapure water/medium-chain triglycerides. In the initial manuscript, the notation γ0 was intended to represent the interfacial tension (γ) value for the system comprising pure water or pure medium-chain triglycerides at a temperature of 25°C. To ensure greater clarity, the relevant section in the revised manuscript has been adjusted, and the modification can be located on lines 161-162.
- In line 197, correct the typo and explain the meaning and equation of ΔT.
Response from the Authors
We sincerely thank the reviewer for their careful and thorough reading of our work. As per the reviewer's suggestion, ΔT represents the time interval used to measure emulsified stability, as established in the majority of relevant references. Given that the time interval in our experiment was consistently 10 minutes, we have substituted ΔT in the formula with the constant value of 10 and provided additional details in line 197 to enhance clarity and understanding.
- Please clarify which samples exhibit significant differences in their mean values (a, b, c, d).
Response from the Authors
We wish to express our sincere gratitude to the reviewer for the valuable comment. In accordance with the reviewer's suggestion, we have augmented the revised manuscript with supplementary explanations regarding the significant differences, specifically within the paragraphs located on lines 285-288 and 405-406.
- The Span concentrations in Table 1 do not match the x-axis in Figure 1.
Response from the Authors
Thank you for pointing the problem. The authors are sorry for our careless mistakes. As suggested by the reviewer, the Table 1 has been corrected in the revised manuscript, as seen in line 101.
- Review the English language, as there are some typographical errors.
Response from the Authors
Thank you very much for your invaluable reminder and constructive suggestions. Taking your insights into consideration, we have meticulously corrected the grammatical errors as mentioned and have thoroughly polished the entire manuscript to ensure its clarity and precision. We are pleased to provide proof of the retouching work done.
- Line 231: Include the cited author’s name.
Response from the Authors
Thank you for pointing the problem. The authors are sorry for our careless mistakes. As suggested by the reviewer, the cited author’s name has been added in the revised manuscript, as seen in line 231.
- Line 266 and others: Correct the unit to cm⁻¹.
Response from the Authors
Thanks to the reviewer for this careful comment. The authors are sorry for our careless mistakes. As suggested by the reviewer, the unit of cm-1 has been corrected in the revised manuscript, as seen in lines 266-271.
- Lines 69-70 refine the text.
Response from the Authors
We appreciate the reviewer's suggestion and have accordingly refined the sentence in lines 69-70. The refined version emphasizes the importance of studying emulsifier-protein interactions in order to enhance the properties and maintain the stability of food emulsions.
- Line 97 include the RPM.
Response from the Authors
Thank you for providing your feedback. We would like to clarify that the content specifically pertaining to RPM, which you suggested needed modification, is not referenced in line 97 of our manuscript. Consequently, we are unable to make an accurate adjustment based on your current suggestion. We sincerely appreciate your insights and would greatly benefit from any more detailed guidance you can provide, which will enable us to make the necessary enhancements and refinements to our article. Please feel free to offer any further suggestions or clarifications.
- Line 142 revise excitation wavelength and emission range.
Response from the Authors
Thank you for this valuable comment. As suggested by the reviewer, the excitation wavelength and emission range has been revised in the manuscript on line 142.
- Line 145: include all dimensions.
Response from the Authors
Thank you for your kind attention to this matter. The authors extend their apologies for any confusion regarding the dimensions of the tablets in the review process. In the original manuscript, the context concerning the size of the tablets indicated that the lyophilized sample was pressed into a disc with a radius of 0.5 cm and a height of 1 mm. To provide a clearer understanding, the corresponding text in the revised manuscript has been reorganized for clarity, and this change can be found on line 145.
- Use a period (.) as the decimal separator instead of a comma (,) for thousands (line 121).
Response from the Authors
Thanks to the reviewer for this careful comment. As suggested by the reviewer, the period has been corrected in the revised manuscript, as seen in line 120.

Reviewer 3 Report
Comments and Suggestions for Authors
Dear authors
Overall a very interesting manuscript. The information should be 'digested' to a a more legible amount by e.g. extracting key information from graphs such as e.g. Fig. 7 and 9 as giving all original measurements makes the graphs illegible and detract from the key information.
Detailed feedback to improve on:
- make sure to stay consistent with the words particle size and droplet size (what particles are meant e.g. in line 16, what is a particle, what a droplet in section 3.1)
- sometimes sentences are not complete, e.g. .line 27/28, 37/38, 57, 70 -> proof reading is suggested
- section 2.1: please give more details on the raw material quality
- Tab. 1: please explain abbreviations as Tables and Figures plus their captions need to be self-explanatory without the surrounding text
- Section 2.15: unclear how this measurement results in a value for emulsion stability. Also, an emulsion stability in minutes is strange (how much of the emulsion is still intact after x minutes? 100%?)
- section 2.17: at what concentration was the foaming done?
- Fig. 4 is illegible in B/W
- section 3.5: please check the discussion - contact angles below 90° should all be hydrophilic and the closer to 90° you get, the less hydrophilic your product should be, so talking about enhanced hydrophobicity for the values you have seems incorrect as hydrophobic particles have contact angles larger than 90°. Further, contact angles close to 90° are good to stabilize emulsions, with angles smaller than 90° good for O/W (and foams), higher than 90 ° for W/O emulsions
- Fig. 5: part of the text not legible
- Fig. 6: scale bars not legible
- Fig. 7 and 9: find way to extract key information and present that instead of the original measurements. These can go to the annex.
- Fig. 10: illegible and hard to interpret. Why not analyse droplet sizes and compare median droplet sizes and X90/X10?
- Fig. 11: find good quantifiable measurement for foaming ability, e.g. how much of the dispersion is foamed and how stable is the foam over time?
- Fig. 12: unclear where sample is taken from - I guess from the foamed layer. Please write so.
- Fig. 12.2 ->Y very interesting but illegible due to too small content
Author Response
- make sure to stay consistent with the words particle size and droplet size (what particles are meant e.g. in line 16, what is a particle, what a droplet in section 3.1)
Response from the Authors
We sincerely thank the reviewer for bringing this issue to our attention. We apologize for any oversights in our manuscript and acknowledge the reviewer's valuable suggestion. As recommended, we have corrected the terminology throughout section 3.1, replacing " droplet size " with "particle size" where applicable in the revised manuscript.
- sometimes sentences are not complete, e.g. .line 27/28, 37/38, 57, 70 -> proof reading is suggested
Response from the Authors
Thank you for your polite reminder. As per the reviewer's suggestion, the sentences have been revised and supplemented to ensure completeness on lines 26-27, 35-37, 57-59, and 70.
- section 2.1: please give more details on the raw material quality
Response from the Authors
Thank you for this careful comment. As suggested by the reviewer, the more details on the raw material quality have been added in the revised manuscript on lines 78-79.
4.Tab. 1: please explain abbreviations as Tables and Figures plus their captions need to be self-explanatory without the surrounding text
Response from the Authors
We thank the reviewer for pointing out this potential confound. To address the reviewer's concern, I would like to clarify the contents of Table 1. This table provides detailed information on the ratios of EYGs maintained at a constant concentration, mixed with varying concentrations of Span emulsifiers. To uniquely identify each sample, a coding system has been established, which employs the initials of EYGs ("E") and Span ("S"), followed by the specific numerical identifiers representing the different types of Span emulsifiers used (20, 40, 60, 80). This coding approach ensures that the table is self-explanatory and facilitates easy reference in subsequent experimental result discussions.
5.Section 2.15: unclear how this measurement results in a value for emulsion stability. Also, an emulsion stability in minutes is strange (how much of the emulsion is still intact after x minutes? 100%?)
Response from the Authors
Thanks for kindly reminding us to clarify this point. The method for assessing emulsion stability, which involves measuring the change in turbidity (through absorbance readings at 500 nm) of an emulsion-SDS mixture over a specified time period (0 and 10 minutes), and calculating the emulsification stability (ES) value. This approach has been documented in numerous studies as a reliable and effective means of evaluating the stability of emulsions. For instance, Liu et al.[2], Meenmanee S et al.[3], and Liu et al.[4] have also utilized similar methodologies in their research to assess emulsion stability. Thus, the method is considered to be highly versatile and applicable to a wide range of emulsion systems.
6.section 2.17: at what concentration was the foaming done?
Response from the Authors
Thank you for your kind reminding us to clarify this point. As depicted in Figure 11.1, this study measured the foam performance of 1% EYGs solution as a control and EYGs-Span solutions, incorporating Span concentrations of 0.001%, 0.01%, and 0.1%.
- Fig. 4 is illegible in B/W
Response from the Authors
Thanks to the reviewer for this careful comment. We would like to clarify that the specific content related to B/W, which you mentioned needing modification, is not referenced in Fig. 4 of our manuscript. Perhaps we guess that you mean that the trend of Fig. 4 is not obvious, so we have enlarged the maximum fluorescence intensity of Fig. 4 in our reply, so that it can be better understood and recognized.
- section 3.5: please check the discussion - contact angles below 90° should all be hydrophilic and the closer to 90° you get, the less hydrophilic your product should be, so talking about enhanced hydrophobicity for the values you have seems incorrect as hydrophobic particles have contact angles larger than 90°. Further, contact angles close to 90° are good to stabilize emulsions, with angles smaller than 90° good for O/W (and foams), higher than 90 ° for W/O emulsions
Response from the Authors
We are grateful for the chance to offer additional clarification on this crucial matter. Several studies have yielded comparable data regarding the contact angle of EYGs, aligning with our findings where the value ranges between 20-40 °, significantly less than 90 °, indicative of hydrophilicity[5, 6]. Upon incorporating the Span emulsifier, we observed a gradual increase in the contact angle approaching 90°, prompting us to designate EYGs as the control group. Notably, the hydrophobicity of the EYGs-Span complex is significantly enhanced compared to natural EYGs, leading us to hypothesize that this complex may augment emulsion stability and improve foaming performance.
- Fig. 5: part of the text not legible
Response from the Authors
We thank the reviewer for pointing out this potential confound. Given the constraints on the size of Figure 5, which integrates both the contact angle and surface hydrophobicity, the depiction of the contact angle has necessarily been scaled down. To facilitate better comprehension, we will furnish a composite figure where the contact angle is represented in a non-photographic format.
Figure 5. Contact angle and surface hydrophobicity of EYGs-Span composite systems with different compounding concentrations. Different letters indicate significant differences (P < 0.05) in the mean values within the same parameter group
10.Fig. 6: scale bars not legible
Response from the Authors
Thanks to the reviewer for this careful comment. Due to the slightly excessive number of spliced pictures, the scale we have indicated appears somewhat indistinct. However, we have clarified in the accompanying note that the scale represents 20μm on line 367.
11.Fig. 7 and 9: find way to extract key information and present that instead of the original measurements. These can go to the annex.
Response from the Authors
Thank you for this valuable comment. In our analysis, Figure 7 presents the dynamic interfacial tension variations over 2400 seconds, plotted with 2400 data points per sample curve, resulting in a dense scatter plot that somewhat obscures trends due to the abundance of samples. Despite our efforts to refine the presentation, the large number of samples and images has necessitated a dense layout. Additionally, Figure 9 combines the emulsification activity and emulsification stability, while we suspect you might be referring to Figure 8 as the diagram depicting the change in composite modulus over time, which similarly aims to capture overall trends by integrating the changes in interface properties under gas-water and oil-water conditions at concentrations of 0.1%, 0.01%, and 0.001%. These data are pivotal to our study as dynamic interfacial tension and interfacial composite modulus are crucial indicators closely related to emulsification and foaming. Importantly, given their significance, these data cannot be included in supplementary materials.
- Fig. 10: illegible and hard to interpret. Why not analyse droplet sizes and compare median droplet sizes and X90/X10?
Response from the Authors
We would like to thank the reviewer for pointing out this issue. Prior to the review process, we had already considered the suggestion of utilizing Image J software for droplet quantification in Figure 10, which showcases a stained emulsion captured under a fluorescence microscope. In this figure, the green areas represent oil droplets and clearly illustrate their overall dispersion. However, despite our attempts to implement this suggestion, we encountered challenges in obtaining accurate results due to the unclear boundaries and overlap of the droplets in the image. For our study, we chose an X40 magnification to better show the overall aggregation and dispersion of droplets, as higher magnifications reduce the number of droplets visible in the field of view.
In response to this issue, we will be including the original, stitched version of Figure 10 in the attachments for your comprehensive review and consideration.
- Fig. 11: find good quantifiable measurement for foaming ability, e.g. how much of the dispersion is foamed and how stable is the foam over time?
Response from the Authors
Thanks to the reviewer for these constructive comments. In generally, the foaming ability (FA) and foaming stability (FS) of the samples were estimated by calculating the foam volume (V0) at 1 min and the foam volume (Vt) at 30 min by the formula[7]. The centrifuge tube depicted in Figure 11 includes a scale, albeit imprecise, allowing only a rough estimation of bubble height and dispersion volume. The figure displays changes at 0 and 30 minutes, though photos were taken every 10 minutes, the limited time span resulted in subtle trends, prompting us to focus on the 0 and 30-minute comparisons for clarity. Furthermore, to visually illustrate how the addition of Span significantly enhances the foaming ability of EYGs, we opted to use images instead of calculations.
14.Fig. 12: unclear where sample is taken from - I guess from the foamed layer. Please write so.
Response from the Authors
Thank you for drawing our attention to this. We appreciate your feedback and clarify that Figure 12 indeed presents the correlation analysis and mechanism diagram, rather than the individual bubble property analysis. We suspect that you might have been referring to Figure 11 instead. We agree that your suggestion enhances clarity in the description of the description of the sample taking process and have accordingly updated the text in section 2.17.
15.Fig. 12.2 ->Y very interesting but illegible due to too small content
Response from the Authors
We are deeply appreciative of the opportunity to clarify this crucial matter further. Upon reviewing Figure 12.2, it is apparent that at lower concentrations, although a modest increase in the adsorption rate of a subset of the EYGs-Span complexes was noted, a larger quantity of native EYGs adsorbed to the interface. As the concentration rose, however, the structure became more relaxed, causing a reduction in the average particle size of the EYGs-Span complexes. The length of the carbon chain in these complexes played a pivotal role: shorter chains led to a looser EYGs structure and a smaller average particle size, facilitating a higher diffusion rate at the interface. Conversely, longer chains resulted in a weakened binding effect. This decrease in steric hindrance, in turn, boosted the adsorption rate, resulting in greater adsorption at the interface and consequently enhancing emulsion dispersion and foam stability.
Please note that due to the constraints of the revision timeline, we were unable to include additional detailed content. However, we acknowledge this limitation and will endeavor to address it in our future endeavors, ensuring continuous improvement in our work.

Reviewer 4 Report
Comments and Suggestions for Authors
I read the manuscript concerning the interfacial properties which play a crucial role in the stability of emulsions and foams. The study investigates how egg yolk granules interact with different Span emulsifiers (Span20, 40, 60, 80) to determine their solution characteristics, interfacial dynamics, and impacts on emulsifying and foaming stability. Findings showed that increasing Span concentrations reduced particle size, raised zeta potential, and led to a looser structure in egg yolk-Span complexes. These structural changes led to lower interfacial tension, higher adsorption rates, and improved composite modulus compared to native egg yolk. The best effects were observed with shorter hydrocarbon chain Spans, while longer chains showed diminished effects. Enhanced interfacial properties resulted in superior emulsifying and foaming stability, with optimal performance achieved in the order of Span20, Span40, Span60, and Span80 as concentrations increased. Correlation analysis indicated a positive relationship between emulsifying stability and interfacial modulus, and a negative relationship with particle size.
The manuscript is pertinent, and the conclusions align well with the arguments put forth. It is well-written, with clear and easily understandable text.
However, before I agree with the publication of the manuscript in Foods, a minor modifications must be done.
1. Authors are asked to enlatge Figures 7, 8 and 12.2 for as better visualization.
2. Authors are asked to introduce in the end of Conclusions much more details about future perspectives regarding the field addressed in the manuscript.
Author Response
- Authors are asked to enlatge Figures 7, 8 and 12.2 for as better visualization.
Response from the Authors
Thanks to the reviewer for this careful comment. As suggested by the reviewer, the Figures 7, 8 and 12.2 have been enlarged in the revised manuscript.
- Authors are asked to introduce in the end of Conclusions much more details about future perspectives regarding the field addressed in the manuscript.
Response from the Authors
Thank you for the insightful feedback. In response to the reviewer's suggestion, we have incorporated detailed future research directions into the conclusions of the revised manuscript, specifically within lines 560-567, providing a richer outlook for the field.

Round 2
Reviewer 2 Report
Comments and Suggestions for Authors
Dear Authors, thank you very much for your reply.
I leave you only the important items that you should continue to work on in order not to confuse with concepts and the quality of the publication.
Regarding item 4,
Sorry for insisting, but I still cannot fully elucidate the methodology and mathematical analysis, including the applied fitting, used to assign the percentages of β-folding, irregular curling, α-helix, and β-turning from the FTIR spectra. Please provide at least an example of an amide I band fitted using peak fit software to help the reader understand the origin of the reported values.
Regarding ítems 5 and 6,
Sorry for insisting on this point, but I would like you to explain in greater detail to the readers and to me the origin of the increase in fluorescence and absorbance and its correlation with the changes in protein conformation.
The authors claim in line 312 that “The UV absorption peak appeared around 280 nm, suggesting that these changes are primarily associated with tyrosine residues“. However, it is observed that increasing the concentration of Span increases the absorbance of the spectrum across the entire wavelength range reported by the authors.
In line 316 authors claim that "These differences may stem from the unfolding and reorganization of EYGs in conjunction with the Span emulsifier at the interface, resulting in enhanced flexibility and increased exposure of chromogenic groups”. I am not convinced that this is solely due to the exposure of chromophores, as the entire spectrum of figure 4B increases, not just a specific band. This suggests that the spectra are strongly affected by light scattering, caused by an increase in emulsification capacity, which leads to smaller particles due to the shorter and more concentrated Span chains. Based on this, it is not clear to me that the increase in the absorbance maximum value at 280 nm is solely attributed to greater light absorption by tryptophan and tyrosine or to increased exposure of tyrosine groups. To improve the interpretation of the results demonstrate that light scattering does not affect the results. Why exposure of tyr due to protein changes could increase absorption in ca. 0.7 au? it is indeed very large, this effect is in general very small.
The same light scattering effect could also be manifesting in the fluorescence results. It would be ideal to show a larger wavelength range to ensure that the data presented are analyzed in a way that light scattering does not affect the results and their subsequent interpretation. Specifically, if tyrosine or tryptophan are exposed to a chemically different environment, this could be manifested as spectral shift, which is not clearly observed in the spectrum due to signal noise.
And regarding item 12,
Again, itis not clear in the Figure 12 if the scheme includes all types of Span in the drawings, with the only effect of concentration. In this sens, it is not easy to include your sentence in the replay nor in the Figure: The shorter the carbon chain of these complexes, the looser the structure of EYGs and the smaller their average particle size, which enhances the diffusion rate at the interface. Meanwhile, the longer the carbon chain, the weaker the binding effect becomes
Author Response
- Regarding item 4,
Sorry for insisting, but I still cannot fully elucidate the methodology and mathematical analysis, including the applied fitting, used to assign the percentages of β-folding, irregular curling, α-helix, and β-turning from the FTIR spectra. Please provide at least an example of an amide I band fitted using peak fit software to help the reader understand the origin of the reported values.
Response from the Authors
Thanks for kindly reminding us to clarify this point. Studies have shown that the amide I band in the Fourier Transform Infrared (FTIR) spectra of protein samples has been proven to be fundamentally a composite of C=O stretching signals from distinct secondary structures within the protein[1]. Utilizing the peak fit software, we can employ three mathematical resolution enhancement techniques: Fourier autoed convolution, second derivative analysis, and band curve fitting[2]. These techniques enable us to dissect and quantify the individual secondary structures within the complex amide I band, thereby revealing the protein's secondary structure through precise calculations[3]. Firstly, the FTIR spectra underwent baseline calibration and smoothing using Omnic software to procure the necessary data. Subsequently, these refined data were imported into peak Fit. In peak Fit, the X-axis region from 1600 to 1700 was selected for baseline calibration and deconvolution fitting, and the second-order derivative was performed to obtain the area and peak associated with β-fold, irregular coil, α-helix and β-turn. Finally, the corresponding area ratio was calculated according to the formula. For better illustration, here is an example image using the peak fit software.
- Regarding ítems 5 and 6,
Sorry for insisting on this point, but I would like you to explain in greater detail to the readers and to me the origin of the increase in fluorescence and absorbance and its correlation with the changes in protein conformation.
The authors claim in line 312 that “The UV absorption peak appeared around 280 nm, suggesting that these changes are primarily associated with tyrosine residues“. However, it is observed that increasing the concentration of Span increases the absorbance of the spectrum across the entire wavelength range reported by the authors.
In line 316 authors claim that "These differences may stem from the unfolding and reorganization of EYGs in conjunction with the Span emulsifier at the interface, resulting in enhanced flexibility and increased exposure of chromogenic groups”. I am not convinced that this is solely due to the exposure of chromophores, as the entire spectrum of figure 4B increases, not just a specific band. This suggests that the spectra are strongly affected by light scattering, caused by an increase in emulsification capacity, which leads to smaller particles due to the shorter and more concentrated Span chains. Based on this, it is not clear to me that the increase in the absorbance maximum value at 280 nm is solely attributed to greater light absorption by tryptophan and tyrosine or to increased exposure of tyrosine groups. To improve the interpretation of the results demonstrate that light scattering does not affect the results. Why exposure of tyr due to protein changes could increase absorption in ca. 0.7 au? it is indeed very large, this effect is in general very small.
The same light scattering effect could also be manifesting in the fluorescence results. It would be ideal to show a larger wavelength range to ensure that the data presented are analyzed in a way that light scattering does not affect the results and their subsequent interpretation. Specifically, if tyrosine or tryptophan are exposed to a chemically different environment, this could be manifested as spectral shift, which is not clearly observed in the spectrum due to signal noise.
Response from the Authors
We are profoundly grateful for the opportunity to further elucidate this pivotal matter. Regarding the genesis of the enhanced fluorescence and absorbance, and their interplay with alterations in protein conformation, Wang et al. [4]and Li et al.[5]have demonstrated that an augmentation in fluorescence intensity serves as an indicator of protein structure unfolding, revealing previously concealed aromatic amino acid residues on the protein's surface and an elevated polarity within the microenvironment. Furthermore, Gou et al.[6]have observed that the exposure of buried hydrophobic groups or the transfer of protein secondary structures to the surface results in increased UV absorption. These findings collectively suggest that the surge in both fluorescence intensity and UV absorbance is intricately linked to the unfolding of the protein structure and the exposure of hydrophobic groups. This correlation is further substantiated by the SEM and FTIR secondary structure analysis of the protein conducted in our study.
The increase in absorbance and fluorescence intensity across the entire wavelength range of the spectrum, observed upon elevating the concentration of Span, can be attributed potentially to non-covalent interactions occurring between the carbonyl oxygen within the EYGs-span complex protein and the hydroxyl hydrogen of the emulsifier. These interactions may induce structural unfolding of the protein, leading to the exposure of hydrophobic groups. Furthermore, as the concentration of Span rises, so does the intensity of these interactions, a phenomenon that can be deduced from the alteration in hydrogen bond characteristic peak 1 evident in the FTIR spectrum[7].
Several studies have indicated that absorbance above 300nm is primarily due to light scattering[8]. In our study, however, the UV spectrum did not exhibit prominent peaks beyond 300nm, prompting us to overlook the potential contribution of light scattering. Regarding the fluorescence spectrum, for the purpose of clearer comparison among samples, we initially presented only the wavelength range of 300-400nm, as our experimental setup had scanned up to 450nm but focused on this narrower range for emphasis. Upon request, we are pleased to provide the fluorescence spectrum covering the entire 300-450nm range for a comprehensive review.
We deeply appreciate the reviewer's insightful question, which has prompted us to delve deeper into this issue. In our subsequent research endeavors, we will give careful consideration to the impact of light scattering. This will undoubtedly guide our future experimental designs and analyses.
- And regarding item 12,
Again, itis not clear in the Figure 12 if the scheme includes all types of Span in the drawings, with the only effect of concentration. In this sens, it is not easy to include your sentence in the replay nor in the Figure: The shorter the carbon chain of these complexes, the looser the structure of EYGs and the smaller their average particle size, which enhances the diffusion rate at the interface. Meanwhile, the longer the carbon chain, the weaker the binding effect becomes.
Response from the Authors
We are deeply appreciative of the opportunity to clarify this crucial matter further. Upon reviewing Figure 12.2, it is apparent that at lower concentrations, although a modest increase in the adsorption rate of a subset of the EYGs-Span complexes was noted, a larger quantity of native EYGs adsorbed to the interface. As the concentration rose, however, the structure became more relaxed, causing a reduction in the average particle size of the EYGs-Span complexes. The length of the carbon chain in these complexes played a pivotal role: shorter chains led to a looser EYGs structure and a smaller average particle size, facilitating a higher diffusion rate at the interface. Conversely, longer chains resulted in a weakened binding effect. This decrease in steric hindrance, in turn, boosted the adsorption rate, resulting in greater adsorption at the interface and consequently enhancing emulsion dispersion and foam stability.
The conclusions presented above are based on the meticulously conducted experimental results in this study. Figure 12.2 serves as a primary schematic diagram of the mechanism, and the unique attributes of the Span emulsifier are also explained in the figure. Given the complexity of visually representing these distinctions and the need to maintain clarity and conciseness in scientific communication, a detailed textual explanation has been prioritized. While we recognize that additional content could provide further insight, we have chosen to focus on delivering a clear and concise message within the constraints of this communication format. However, we acknowledge this limitation and will endeavor to address it in our future endeavors, ensuring continuous improvement in our work.

Reviewer 3 Report
Comments and Suggestions for Authors
Dear authors
Thank you for claryifing and editing where possible. I understand your constraints and would for a next time still suggest to go for less dense pictures as the legibility would improve but find it acceptable with your explanations.
The content is highly valuable - congratulations on this nice and extensive work.
Author Response
Dear authors
Thank you for claryifing and editing where possible. I understand your constraints and would for a next time still suggest to go for less dense pictures as the legibility would improve but find it acceptable with your explanations.
The content is highly valuable - congratulations on this nice and extensive work.
Response from the Authors
We deeply appreciate your concise summary of the manuscript and your encouraging commentary. We extend our heartfelt gratitude for your diligent and professional review work, your insightful and constructive comments, and your invaluable suggestions pertaining to our manuscript. Your dedication of time and effort is sincerely esteemed and greatly cherished.
